# MMaDA:
# Multimodal Large Diffusion Language Models

**Ling Yang**[1,4*†], **Ye Tian**[2*], **Bowen Li**[2], **Xinchen Zhang**[3],
**Ke Shen,**[4] **Yunhai Tong**[2], **Mengdi Wang**[1†]
[1]Princeton University  [2]Peking University [3]Tsinghua University [4]ByteDance Seed

## Abstract

We introduce MMaDA, a novel class of multimodal diffusion foundation models designed to achieve superior performance across diverse domains such as textual reasoning, multimodal understanding, and text-to-image generation. The approach is distinguished by three key innovations: **(i)** MMaDA adopts a unified diffusion architecture with a shared probabilistic formulation and a modality-agnostic design, eliminating the need for modality-specific components. This architecture ensures seamless integration and processing across different data types. **(ii)** We implement a *mixed long chain-of-thought (CoT) fine-tuning* strategy that curates a unified CoT format across modalities. By aligning reasoning processes between textual and visual domains, this strategy facilitates cold-start training for the final reinforcement learning (RL) stage, thereby enhancing the model's ability to handle complex tasks from the outset. **(iii)** We propose *UniGRPO*, a unified policy-gradient-based RL algorithm specifically tailored for diffusion foundation models. Utilizing diversified reward modeling, *UniGRPO* unifies post-training across both reasoning and generation tasks, ensuring consistent performance improvements. Experimental results demonstrate that **MMaDA-8B** exhibits strong generalization capabilities as a unified multimodal foundation model. It surpasses powerful models like LLaMA-3-7B and Qwen2-7B in textual reasoning, outperforms Show-o and SEED-X in multimodal understanding, and excels over SDXL and Janus in text-to-image generation. These achievements highlight MMaDA 's effectiveness in bridging the gap between pretraining and post-training within unified diffusion architectures, providing a comprehensive framework for future research and development. We open-source our code and trained models at: `https://github.com/Gen-Verse/MMaDA`

## 1 Introduction

Large language models (LLMs) have revolutionized natural language processing (NLP) by achieving state-of-the-art performance in diverse tasks, from text generation (e.g., ChatGPT [1–3]) to complex reasoning (e.g., DeepSeek-R1 [4]). Inspired by their success, the research community has extended LLMs to the multimodal domain, giving rise to multimodal large language models (MLLMs) or vision-language models (VLMs) [5–14], such as GPT-4 [15] and Gemini [12]. These models aim to provide a unified framework for both understanding and generating across heterogeneous modalities—text, images, and beyond.

Early multimodal approaches combined language models with diffusion models [16–19] to handle discrete (e.g., text) and continuous (e.g., image) modalities separately. Subsequent autoregressive (AR) methods simplified architectures by training a single transformer with next-token prediction,

---

*Equal Contribution.

†Corresponding Authors. Contact: yangling0818@163.com, mengdiw@princeton.edu

39th Conference on Neural Information Processing Systems (NeurIPS 2025).

unifying discrete and continuous generation in a single model [12, 8, 9, 11, 10, 13, 14]. Another line of work leverages modality-specific training objectives within a shared architecture: for example, Show-o [20] and Transfusion [21] combine autoregressive and diffusion modeling for modeling textual and visual semantics, respectively.

Table 1: Specific design choices employed by different unified multimodal foundation model families, including their core loss functions. The next-token prediction loss is defined as $\mathcal{L}_{\text{NTP}} = \mathbb{E}_{x_i} \left[ -\log P_\theta(x_i \mid x_{<i}) \right]$, representing the standard negative log-likelihood of generating the next token $x_i$ conditioned on its preceding context $x_{<i}$. The training objective for continuous diffusion models is given by $\mathcal{L}_{\text{Diff-cont}} = \mathbb{E}_{t, x_0 \sim q(x_0), \epsilon \sim \mathcal{N}(\mathbf{0}, \mathbf{I}), c} \left[ \|\epsilon - \epsilon_\theta(x_t, t, c)\|^2 \right]$, where $x_t$ denotes the noised version of the original data $x_0$ at timestep $t$, and $c$ is an optional conditioning signal. The model learns to predict the noise $\epsilon$ added to the data, thereby enabling the reconstruction of $\mathbf{x}_0$ through an iterative denoising process defined as $x_{t-1} = \mathcal{F}_\theta(x_t, t)$. The discrete diffusion (masked token prediction) loss is given by $\mathcal{L}_{\text{Diff-disc}} = \mathbb{E}_{z_i^*} \left[ -\frac{1}{|M|} \sum_{i \in M} \log p_\theta(z_i^* | \mathbf{z}_{\text{masked}}^{(M)}) \right]$, which measures the average negative log-likelihood of correctly predicting the original discrete tokens $z_i^*$ at positions masked by the set $M$, given the visible context provided by the masked sequence $\mathbf{z}_{\text{masked}}^{(M)}$.

| | **AR** (One Model) | **AR + Diffusion** (Two Models) | **AR + Diffusion** (One Model) | **Ours** MMaDA (One Model) |
|---|---|---|---|---|
| **Network Architecture** | | | | |
| Language | AR | AR | AR | Diffusion |
| Vision | AR | Diffusion | Diffusion | Diffusion |
| **Pre-training** | | | | |
| Loss for Language | $\mathcal{L}_{\text{NTP}}$ | $\mathcal{L}_{\text{NTP}}$ | $\mathcal{L}_{\text{NTP}}$ | $\mathcal{L}_{\text{Diff-disc}}$ |
| Loss for Vision | $\mathcal{L}_{\text{NTP}}$ | $\mathcal{L}_{\text{Diff-cont}}$ | $\mathcal{L}_{\text{Diff-cont}}$ or $\mathcal{L}_{\text{Diff-disc}}$ | $\mathcal{L}_{\text{Diff-disc}}$ |
| **Sampling** | | | | |
| Language | $x_i \sim P_\theta(x_i \mid x_{<i})$ | $x_i \sim P_\theta(x_i \mid x_{<i})$ | $x_i \sim P_\theta(x_i \mid x_{<i})$ | $x_{t-1} = \mathcal{F}_\theta(x_t, t)$ |
| Vision | $x_i \sim P_\theta(x_i \mid x_{<i})$ | $x_{t-1} = \mathcal{F}_\theta(x_t, t)$ | $x_{t-1} = \mathcal{F}_\theta(x_t, t)$ | $x_{t-1} = \mathcal{F}_\theta(x_t, t)$ |
| Language Scheduler | - | - | - | Semi-AR Remask [22] |
| Vision Scheduler | - | DDPM [23] | DDPM [23]/Cosine [24] | Cosine Remask [24] |
| **Post-Training** | | | | |
| Language CoT | - | - | - | Mixed Long-CoT |
| Language RL | - | - | - | UniGRPO with Diversified RM |
| Vision CoT | - | - | - | Mixed Long-CoT |
| Vision RL | DPO [25] | - | - | UniGRPO with Diversified RM |
| **Tasks** | | | | |
| Und. (with Reasoning) | ✓(✗) | ✓(✗) | ✓(✗) | ✓(✓) |
| Image Gen. (with Reasoning) | ✓(✗) | ✓(✗) | ✓(✗) | ✓(✓) |
| Text Gen. (with Reasoning) | ✗(✗) | ✗(✗) | ✗(✗) | ✓(✓) |
| **Representative Models** | Emu3 [25], Janus [13, 14] | DreamLLM [16] | Transfusion [21], Show-o [20] | MMaDA (Ours) |

Although recent advancements have explored diffusion-based architectures for global context modeling and parallel generation [20, 21], existing unified multimodal foundation models predominantly focus on model architecture design and pretraining strategies, leaving a critical gap in the exploration of **post-training methodologies**, particularly in **non-autoregressive settings**.

To address this gap, we systematically investigate the design space for unified multimodal diffusion foundation models, introducing a novel framework that advances both architectural and training paradigms (a comprehensive comparison in Table 1). This work bridges the gap between pretraining and post-training in unified multimodal diffusion models, offering a holistic framework for future research in this emerging field. Our contributions can be summarized as:

- **Unified Diffusion Foundation Architecture:** We propose MMaDA, a class of diffusion-based models that extend traditional generators into generalist task solvers via a *shared probabilistic formulation* and *modality-agnostic architecture*. This design eliminates modality-specific components while maintaining strong performance across tasks.

- **Mixed Long-CoT Post-Training:** We introduce *mixed long chain-of-thought (CoT) finetuning* to enable cold-start training. By curating a unified CoT format across tasks, we align reasoning processes between modalities (e.g., textual and visual), fostering cross-modal synergy and learning intermediate reasoning before final output generation.

- **Unified Reinforcement Learning (UniGRPO):** We develop a *unified diffusion-centric* reinforcement learning algorithm (*UniGRPO*) tailored for multimodal generation. This approach leverages *diversified reward modeling* to enhance the model's ability to perform complex reasoning and maintain factual consistency in generation.

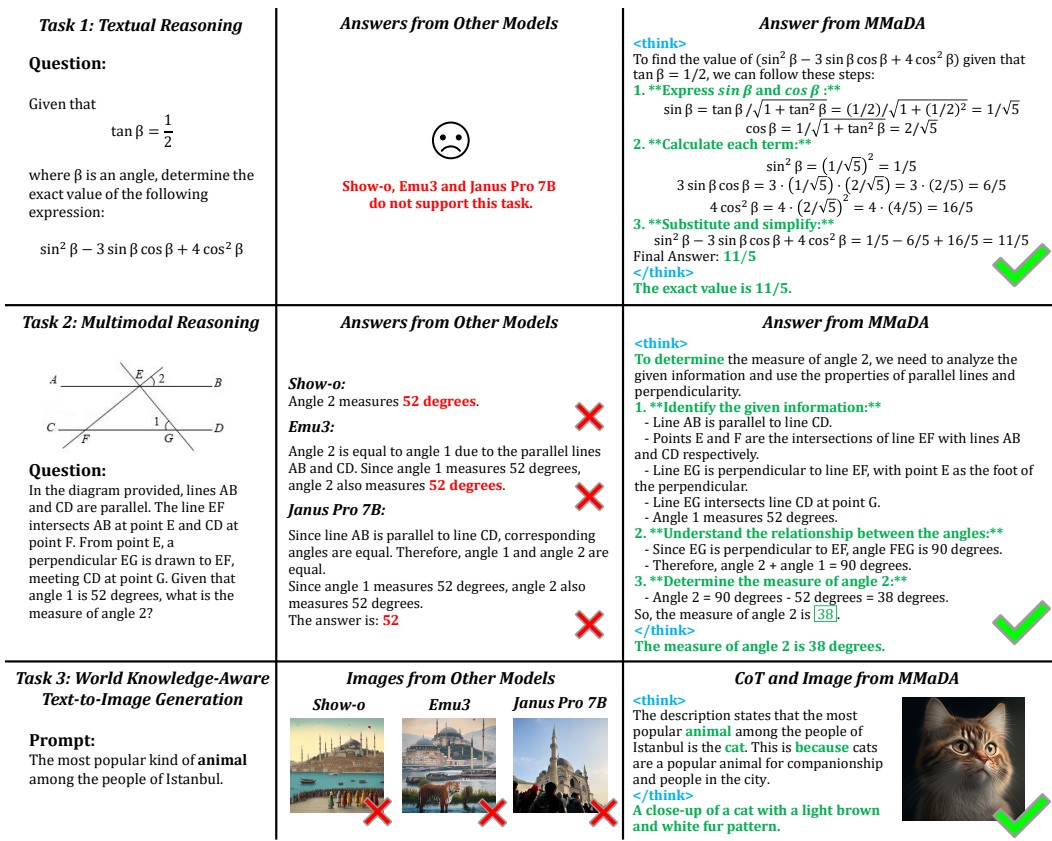

Figure 1: Qualitative comparison across three tasks (more results in Appendix F and Appendix G.).

- **State-of-the-Art Performance:** MMADA achieves superior and balanced performance across three critical tasks: *textual reasoning*, *multimodal understanding*, and *text-to-image generation*. Notably, it outperforms both autoregressive and diffusion-based baselines in terms of accuracy, efficiency, and task adaptability.

## 2   Related Work

Recently, numerous studies [19, 26–28, 16, 29] have focused on developing unified multimodal foundation models that excel in both understanding and generation. Approaches such as SEED-X [19] and DreamLLM [16], along with others [5–14], represent all modalities as a series of tokens and employ a unified transformer architecture to train the entire system end-to-end. For example, Emu3 [25] trains a single transformer from scratch using a mixture of multimodal tokenized sequences, optimized solely with next-token prediction. While these unified autoregressive models show promise, they can struggle with visual generation tasks. Transfusion [21] and Show-o [20] employ autoregressive modeling for text generation and diffusion modeling for visual generation. Similarly, VAR-GPT [30] models visual understanding and generation with autoregressive modeling and visual autoregressive modeling(VAR [31]), respectively. Nevertheless, they mainly focus on pretraining strategies. Exploring effective post-training designs is still lacking for existing unified multimodal foundation models. (More related works can be found in Appendix A)

## 3   MMaDA: Multimodal Large Diffusion Language Models

### 3.1   Pretraining with Unified Diffusion Architecture and Objective

**Data Tokenization**    To establish a unified modeling framework capable of processing both textual and visual data, we adopt a consistent discrete tokenization strategy across both modalities. This design enables the model to operate under a single modeling objective, i.e., the prediction of discrete masked tokens. For text tokenization, we utilize the tokenizer from LLaDA [22], which serves as the backbone for our MMaDA model. For image tokenization, we leverage the pretrained image quantizer adopted from Show-o [20], which is based on the MAGVIT-v2 [24] architecture and converts raw

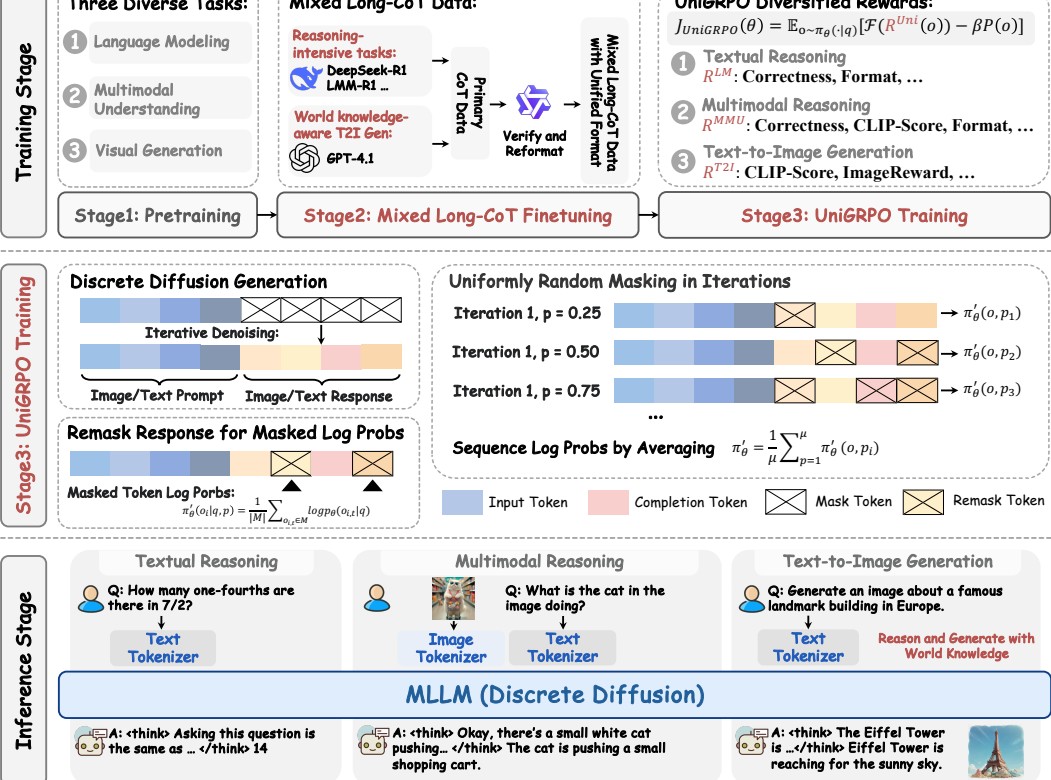

Figure 2: An overview of MMaDA pipeline.

image pixels into sequences of discrete semantic tokens. Given an input image with dimensions $H \times W$, the encoder generates a token map with dimensions $\frac{H}{f} \times \frac{W}{f}$, where $f$ represents the downsampling factor. In this implementation, we employ a downsampling factor of $f = 16$ and a codebook size of 8192. This configuration transforms a $512 \times 512$ pixel image into a sequence of $32 \times 32 = 1024$ discrete tokens. The transformed discrete image tokens are used in both understanding and generation modeling tasks.

**Unified Probabilistic Formulation for Pretraining**    Recent unified multimodal frameworks aim to integrate multiple modeling objectives—such as autoregressive generation and diffusion-based denoising—into a single architecture for joint understanding and generation tasks [20, 21] (see preliminaries in Appendix B.1). However, these approaches often introduce complex hybrid mechanisms that hinder model efficiency and coherence. In contrast, we propose a streamlined framework that not only simplifies the architectural complexity but also introduces a *unified diffusion objective* to model both visual and textual modalities under a shared probabilistic formulation. By aligning the noise corruption and semantic recovery processes across modalities, we enable more effective cross-modal interactions during pretraining, facilitating seamless integration of heterogeneous data sources.

Specifically, we formulate MMaDA as a mask token predictor (for both image and text tokens), a parametric model $p_\theta(\cdot|x_t)$ that takes $x_t$ as input and predicts all masked tokens simultaneously. The model is trained with a unified cross-entropy loss computed only on the masked image/text tokens:

$$\mathcal{L}_{\text{unify}}(\theta) = -\mathbb{E}_{t,x_0,x_t} \left[ \frac{1}{t} \sum_{i=1}^{L} \mathbf{I}[x_t^i = \texttt{[MASK]}] \log p_\theta(x_0^i | x_t) \right], \tag{1}$$

where $x_0$ is ground truth, the timestep $t$ is sampled uniformly from $[0, 1]$, and $x_t$ is obtained by applying the forward diffusion process to $x_0$. $\mathbf{I}[\cdot]$ denotes the indicator function to ensure that the loss is computed only over the masked tokens. Specific pretraining tasks are detailed in Section 4.

## 3.2 Post-Training with Mixed Long-CoT Finetuning

**Cold Start Long-CoT Data Curation**   We investigate how CoT mechanisms [32] can enhance post-training for our unified multimodal diffusion framework and observe their effectiveness in promoting cross-modal synergies. To this end, we curate a compact dataset of long CoT trajectories across three core tasks: *textual reasoning*, *multimodal reasoning*, and *text-to-image generation*. This dataset enables stable post-training of our pretrained MMaDA model through the following principles:

- **Unified CoT Format**: A critical challenge in vision-language foundation models is the heterogeneity of output formats across tasks (e.g., text vs. image generation). We propose a *task-agnostic* CoT format:

    |<special_token>| < reasoning_process > |<special_token>| < result > .

    The "<reasoning_process>" encodes step-by-step reasoning trajectories preceding the final output. This unified structure bridges modality-specific outputs and facilitates knowledge transfer between tasks. For instance, enhanced textual reasoning capabilities directly improve the realism of generated images by aligning semantic logic with visual synthesis.

- **Diversity, Complexity, and Accuracy**: We leverage open-source large language and vision-language models (LLM/VLMs) to generate diverse reasoning trajectories across tasks (in Fig. 2). To ensure quality, we employ state-of-the-art models as *verifiers* to filter out inaccurate or shallow reasoning, selecting only high-quality, long-form CoT samples. Unlike prior unified models that focus on generic understanding and generation, our MMaDA is explicitly designed for:

    1. *Reasoning-intensive tasks* (e.g., mathematical problem-solving), and
    2. *World-knowledge-aware text-to-image generation*, where factual consistency is critical.

**Mixed Long-CoT Finetuning**   Leveraging our unified diffusion architecture and probabilistic formulation, we develop a mixed-task long-CoT finetuning strategy to jointly optimize the model across heterogeneous tasks. This approach not only enhances task-specific capabilities but also creates a strong initialization for subsequent reinforcement learning (RL) stages. The training process follows these steps:

1. **Prompt Preservation and Token Masking**: We retain the original prompt $p_0$ and independently mask tokens in the result ($x_0$), denoted as $r_t$.

2. **Joint Input and Loss Computation**: The concatenated input $[p_0, r_t]$ is fed into our pretrained mask predictor to compute the loss. This enables the model to reconstruct masked regions ($r_0$) using contextual information from both the prompt and corrupted result.

The objective function is defined as:

$$\mathcal{L}_{\text{Mixed-SFT}} = -\mathbb{E}_{t,p_0,r_0,r_t} \left[ \frac{1}{t} \sum_{i=1}^{L'} \mathbf{I}[r_t^i = \texttt{[MASK]}] \log p_\theta(r_0^i | p_0, r_t) \right], \tag{2}$$

where $L'$ denotes the sequence length. Here, $[p_0, r_0]$ and $[p_0, r_t]$ correspond to the clean data $x_0$ and its noisy counterpart $x_t$, respectively. This formulation ensures the model learns to recover masked tokens while maintaining alignment with the original prompt and task-specific reasoning logic.

## 3.3 Post-Training with Unified Reinforcement Learning

### 3.3.1 Unified GRPO for Diffusion Foundation Models

With our mixed long-CoT fine-tuning, MMaDA demonstrates the ability to generate unified and coherent reasoning chains prior to final outputs. To further enhance its performance on **knowledge-intensive tasks** and **complex reasoning/generation scenarios**, we propose **UniGRPO**, a novel policy-gradient-based reinforcement learning algorithm tailored for diffusion foundation models. This approach enables a *diffusion-centric RL training framework* that unifies task-specific objectives across diverse modalities and reasoning paradigms. The method is structured into two core components: (1) a *unified mathematical formulation* for diffusion-based RL, and (2) *diversified reward modeling* to align policy gradients with task-specific rewards.

**Challenges in Adapting Autoregressive GRPO to Diffusion Models**   The original GRPO [33] relies on computing token-level log-likelihoods $\pi_\theta(o_{i,t}|q, o_{i,<t})$ and sequence-level probabilities $\pi_\theta$ and $\pi_{ref}$ (preliminary in Appendix B.2). In autoregressive (AR) LLMs, these metrics are efficiently derived via the chain rule of generation. However, diffusion models introduce three critical challenges: **(1) Local Masking Dependency**: Token-level log-likelihoods $\log \pi_\theta(o_{i,t}|q, o_{i,<t})$ are only valid within masked regions during the diffusion process, unlike AR models where all tokens are valid. **(2) Mask Ratio Sensitivity**: A uniform mask ratio must be sampled for the response segment to approximate the policy distribution $\pi_\theta$, as diffusion dynamics depend on masking patterns. **(3) Non-Autoregressive Sequence-Level Likelihoods**: The sequence-level log-likelihood cannot be directly accumulated from token-level probabilities due to the absence of an autoregressive chain rule in diffusion models. Prior approaches address these issues with suboptimal strategies. LLaDA [22] employs Monte Carlo sampling over numerous mask ratios (e.g., 128 samples), incurring high computational costs for on-policy RL. d1 [34] fixes the mask ratio and randomizes question masking, which reduces noise diversity and ignores the multi-step denoising nature of diffusion.

**Unified Formulation for Diffusion GRPO**   To overcome these limitations, we introduce **Uni-GRPO**, a *computationally efficient approximation algorithm* designed for diffusion architectures. Given a batch of responses $\{o_i\}_{i=1}^G$ for a query $q$, each response is fixed during gradient updates to ensure stable policy evaluation. We highlight three critical points of our UniGRPO as:

1. **Structured Noising Strategy**: For each $o_i$, we sample a masking ratio $p_i \in [0, 1]$ uniformly and construct a perturbed version $\tilde{o}_{i,p}$ by replacing tokens with [MASK]. The random seed for $p_i$ varies across gradient steps, ensuring diverse masking patterns during training.

2. **Efficient Log-Likelihood Approximation**: We define the expected per-token log-likelihood under the perturbed distribution as:

$$\pi_\theta'(o_{i,t} \mid q, \tilde{o}, p_i) = \mathbb{E}_{p_i \sim [0,1]}\left[\mathbf{I}[o_{i,t,p} = \text{[MASK]}] \log p_\theta(o_{i,t,p}|q)\right] \tag{3}$$

   The sequence-level log-likelihood is then approximated by averaging over masked tokens:

$$\pi_\theta' = \frac{1}{\text{M}} \sum_{o_{i,t} \in \text{M}} \log p_\theta(o_{i,t}|q), \quad \text{where M denotes the number of masked tokens.} \tag{4}$$

3. **Policy Gradient Objective**: The per-token reward is computed as the ratio between current and old policy likelihoods: $r_{i,t}'(\theta) = \frac{\pi_\theta'(o_{i,t}|q,\tilde{o},p_i)}{\pi_{\text{old}}'(o_{i,t}|q,\tilde{o},p_i)}$. The final UniGRPO objective integrates clipped surrogate rewards and KL regularization:

$$\mathcal{J}_{\text{UniGRPO}}(\theta) = \mathbb{E}_{(q,a)\sim\mathcal{D}, \{o_i\}_{i=1}^G \sim \pi_{\theta_{\text{old}}}(\cdot|q), \{p_i \in [0,1]\}_{i=1}^G} \left[ \frac{1}{G} \sum_{i=1}^G \frac{1}{|o_i|} \sum_{t=1}^{|o_i|} \left( \min\left(r_{i,t}'(\theta)\hat{A}_{i,t}, \right.\right.\right.$$
$$\left.\left.\left. \text{clip}\left(r_{i,t}'(\theta), 1-\varepsilon, 1+\varepsilon\right)\hat{A}_{i,t}\right) - \beta D_{\text{KL}}(\pi_\theta'||\pi_{\text{ref}}')\right)\right],$$
$$\tag{5}$$

   where $\hat{A}_{i,t}$ denotes the advantage estimate, $\varepsilon$ controls the clipping range, and $\beta$ balances the KL divergence penalty.

For further details on the UniGRPO algorithm, please refer to Appendix D and Appendix E.2.

### 3.3.2   Diversified Reward Modeling

We further simplify the optimization objective of UniGRPO (Eq. (5)) as follows:

> **Remark 1.** *General optimization objective of UniGRPO:*
> $$\mathcal{J}_{\text{UniGRPO}}(\theta) = \mathbb{E}_{o \sim \pi_\theta(\cdot|q)}[\mathcal{F}(R^{\text{Uni}}(o)) - \beta P(o)], \tag{6}$$

where $R^{\text{Uni}}(o)$ denotes the reward obtained from the model-generated response $o$, $P(\cdot)$ is the penalty term, which denotes the KL divergence as specified in Eq. (5). This is a unified rule-based reward system, where $R^{\text{Uni}}(\cdot)$ can be instantiated with diverse rewards for different tasks. To address

Table 2: Evaluation on Multimodal Understanding Benchmarks.

| Model | POPE↑ | MME↑ | Flickr30k↑ | VQAv2$_{(test)}$↑ | GQA↑ | MMMU↑ | MMB↑ | SEED↑ |
|---|---|---|---|---|---|---|---|---|
| *Understanding-Only* | | | | | | | | |
| LLaVA-v1.5 [38] | 85.9 | 1510.7 | - | 78.5 | 62.0 | 35.4 | 64.3 | 58.6 |
| InstructBLIP [39] | 78.9 | 1212.8 | - | - | 49.5 | - | - | - |
| Qwen-VL-Chat [40] | - | 1487.5 | - | 78.2 | 57.5 | - | 60.6 | 58.2 |
| mPLUG-Owl2 [41] | 85.8 | 1450.2 | - | 79.4 | 56.1 | - | - | - |
| LLaVA-Phi [42] | 85.0 | 1335.1 | - | 71.4 | - | - | 59.8 | - |
| *Unified Understanding & Generation* | | | | | | | | |
| DreamLLM [16] | - | - | - | 72.9 | - | - | - | - |
| SEED-X [19] | 84.2 | 1435.7 | 52.3 | - | 47.9 | 35.6 | - | - |
| Chameleon [8] | - | - | 74.7 | 66.0 | - | - | - | - |
| LWM [9] | 75.2 | 948.4 | - | 55.8 | 44.8 | - | - | - |
| Emu [11] | - | - | 77.4 | 57.2 | - | - | - | - |
| Show-o [20] | 80.0 | 1097.2 | 62.5 | 69.4 | 58.0 | 26.7 | - | - |
| Gemini-Nano-1 [12] | - | - | - | 62.7 | - | 26.3 | - | - |
| MMaDA (**Ours**) | 86.1 | 1410.7 | 67.6 | 76.7 | 61.3 | 30.2 | 68.5 | 64.2 |

the varied requirements of different tasks, we have defined a range of rewards under the unified formulation Eq. (6), providing tailored RL optimization directions for each task branch. We mainly adopt three types of rewards:

- **Textual Reasoning Rewards**: We apply UniGRPO on the training split of the GSM8K [35] dataset and define a composite reward. This includes a **Correctness Reward** of 2.0 for a correct answer, and a **Format Reward** of 0.5 if the response adheres to our predefined format: "`<think>...</think>`".

- **Multimodal Reasoning Rewards**: For mathematical tasks such as GeoQA [36] and CLEVR [37], we adopt the same **Correctness** and **Format Rewards** as in textual reasoning. In addition, for caption-based tasks, we further introduce a **CLIP Reward** of $0.1 \cdot \text{CLIP}(\text{image}, \text{text})$, where the original CLIP score measuring text-image alignment is scaled by $0.1$ to balance its influence.

- **Text-to-Image Generation Rewards**: For image generation tasks, we incorporate the same **CLIP Reward** to assess text-image semantic alignment, alongside an **Image Reward** that reflects human preference scores. Both rewards are scaled by a factor of $0.1$ to ensure balanced contribution during optimization.

## 4 Experiments

**Datasets** To train MMaDA, we utilized a diverse range of datasets tailored for corresponding training stages as follows: (1) **Foundational Language and Multimodal Data**: For basic text generation capabilities, we adopt the RefinedWeb [43] dataset. For multimodal understanding and generation tasks, we incorporate image-text datasets including ImageNet-1k [44], CC12M [45], SA1B [46], LAION-aesthetics-12M [47], and JourneyDB [48]. (2) **Instruction Tuning Data**: To enhance instruction-following capabilities, we use Alpaca [49] for textual instructions and LLaVA-1.5 [38] for visual instruction tuning. (3) **Reasoning Data**: For Mixed Long-CoT finetuning, we curated a diverse set of reasoning datasets. For textual mathematical and logical reasoning, we employed LIMO [50], s1k [51], OpenThoughts [52], and AceMath-Instruct [53]. For multimodal reasoning, we used the LMM-R1 [54] model to generate responses on GeoQA [36] and CLEVR [37], and retained correctly answered instances. Additionally, for world knowledge-aware image generation, we used GPT-4.1 to synthesize factual item-description pairs spanning science, culture, and landmarks, formatted into unified CoT-style traces.(4) **Reinforcement Learning Data**: For UniGRPO training, we adopt the original mathematical and logical datasets used in Reasoning [55, 36, 37].

**Evaluation and Baselines** We evaluate our MMaDA on three distinct tasks using task-specific metrics and baselines:(1) **Multimodal Understanding:** Following LLaVA [38], we evaluate on POPE, MME, Flickr30k, VQAv2, GQA, and MMMU, and compare against understanding-only models [38–42], as well as unified models [19, 16, 8, 9, 11, 20, 12]. (2) **Image Generation:** We assess generation

Table 3: Evaluation on Image Generation Benchmarks.

| Model | Wise (Natural)↑ | Image Reward↑ | CLIP Score↑ | GenEval↑ | | | | | | |
|---|---|---|---|---|---|---|---|---|---|---|
| | | | | Single Obj. | Two Obj. | Counting | Colors | Position | Color Attr. | Overall |
| *Generation-Only* | | | | | | | | | | |
| LlamaGen [63] | - | 0.79 | 13.43 | 0.71 | 0.34 | 0.21 | 0.58 | 0.07 | 0.04 | 0.32 |
| SDv1.5 [60] | 0.34 | 0.84 | 23.54 | 0.97 | 0.38 | 0.35 | 0.76 | 0.04 | 0.06 | 0.43 |
| SDv2.1 [60] | 0.30 | 0.95 | 27.41 | 0.98 | 0.51 | 0.44 | 0.85 | 0.07 | 0.17 | 0.50 |
| DALL-E 2 [61] | - | 0.83 | 25.20 | 0.94 | 0.66 | 0.49 | 0.77 | 0.10 | 0.19 | 0.52 |
| SDXL [62] | 0.43 | 1.13 | 32.12 | 0.98 | 0.74 | 0.39 | 0.85 | 0.15 | 0.23 | 0.55 |
| *Unified Understanding & Generation* | | | | | | | | | | |
| DreamLLM [16] | - | 0.76 | 18.33 | - | - | - | - | - | - | - |
| SEED-X [19] | - | 0.77 | 23.15 | 0.97 | 0.58 | 0.26 | 0.80 | 0.19 | 0.14 | 0.49 |
| Chameleon [8] | - | 0.83 | 20.32 | - | - | - | - | - | - | 0.39 |
| LWM [9] | - | 0.78 | 26.21 | 0.93 | 0.41 | 0.46 | 0.79 | 0.09 | 0.15 | 0.47 |
| Emu [11] | - | 0.81 | 22.29 | - | - | - | - | - | - | - |
| Show-o [20] | 0.28 | 0.92 | 28.94 | 0.95 | 0.52 | 0.49 | 0.82 | 0.11 | 0.28 | 0.53 |
| Janus [13] | 0.16 | 1.03 | 29.45 | 0.97 | 0.68 | 0.30 | 0.84 | 0.46 | 0.42 | 0.61 |
| Gemini-Nano-1 [12] | - | 0.89 | 24.58 | - | - | - | - | - | - | - |
| VAR-GPT [30] | - | 0.94 | 28.85 | 0.96 | 0.53 | 0.48 | 0.83 | 0.13 | 0.21 | 0.53 |
| MMaDA (**Ours**) | 0.67 | 1.15 | 32.46 | 0.99 | 0.76 | 0.61 | 0.84 | 0.20 | 0.37 | 0.63 |

quality using 50K prompts from our test set to compute CLIP Score [56] and ImageReward [57] to evaluate textual alignment and human preference alignment. We adopt GenEval [58] for general evaluation and WISE [59] for evaluating world knowledge-based generation, comparing against generation-specific models [60–63] and unified baselines [19, 16, 8, 9, 11, 13, 20, 12, 30]. (3) **Text Generation:** we evaluate instruction-following and reasoning performance on MMLU, GSM8K, and related benchmarks, comparing with LLaMA2-7B, Qwen2-7B, LLaDA-8B and Dream-7B [64]. We detail our inference mechanism for text and image generation in Appendix C.

**Implementation Details** We initialize MMaDA with LLaDA-8B-Instruct's pretrained weights [22] and an image tokenizer with Show-o's pretrained ones. We perform joint training across three stages: **Stage1:** The initial model is trained for 200K steps using foundational language and multimodal data, including RefinedWeb for text generation, ImageNet-1k for class-conditional image generation, and additional image-text datasets for captioning. This is followed by another 400K steps where ImageNet is replaced with more diverse image-text pairs. **Stage2:** The model is then jointly trained for 50,000 steps using Instruction Tuning Data and Reasoning Data. **Stage3:** This final stage consists of UniGRPO training with Reinforcement Learning Data for 50,000 steps. Training is performed on 64 A100 (80GB) GPUs using a global batch size of 1,280. The AdamW optimizer is employed with an initial learning rate of 5e-5 and a cosine learning rate scheduler.

**Multimodal Understanding** Table 2 reports the multimodal understanding performance of our method on standard benchmarks, including POPE, MME, Flickr30k, VQAv2, GQA, and MMMU. For outputs from MMaDA that contain reasoning traces, we use the final answer as the prediction. Compared with dedicated understanding-only models such as LLaVA-v1.5, InstructBLIP, and Qwen-VL-Chat, our model achieves comparable or superior results across most benchmarks, despite being trained under a unified objective. When compared to other unified models (e.g., SEED-X, DreamLLM, Janus, Emu3, and Show-o), our method consistently outperforms them across several benchmarks, particularly benefiting from the proposed Mixed Long-CoT Finetuning and UniGRPO Reinforcement Learning stages. Notably, this is the first demonstration of a diffusion-based MLLM exhibiting strong understanding capabilities, highlighting the potential of our unified architecture in bridging generation and understanding tasks. Qualitative results are in Appendix F and Appendix G.

**Text-to-Image Generation** Table 3 presents the evaluation results on text-to-image generation benchmarks. Our model achieves the highest performance in both CLIP Score and ImageReward across generation-only and unified models, attributed to the UniGRPO training stage with rewards explicitly aligned to these metrics. Furthermore, our method demonstrates superior compositionality and object counting capabilities on GenEval, benefiting from the reasoning-intensive training of the understanding branch. Notably, on WISE [59] Natural benchmark, which is designed to evaluate world knowledge-aware generation, our model significantly outperforms prior approaches, owing to its joint training on text-based reasoning, which is typically absent in existing unified models. Qualitative results are in Appendix F and Appendix G.

Table 4: Evaluation on LLM Benchmarks.

| Model | Arch | MMLU | ARC-C | TruthfulQA | GSM8K | MATH | GPQA |
|-------|------|------|-------|------------|-------|------|------|
| LLaMA2-7B | AR | 45.9 | 46.3 | 39.0 | 14.3 | 3.2 | 25.7 |
| LLaMA3-8B | AR | 64.5 | 53.1 | 44.0 | 53.1 | 15.1 | 25.9 |
| Qwen2-7B | AR | 70.3 | 60.6 | 54.2 | 80.2 | 43.5 | 30.8 |
| LLaDA-8B | Diffusion | 65.9 | 47.9 | 46.4 | 70.7 | 27.3 | 26.1 |
| Dream-7B | Diffusion | 67.0 | - | - | 81.0 | 39.2 | 33.0 |
| MMaDA-8B (**Ours**) | Diffusion | 68.4 | 57.4 | 43.1 | 73.4 | 36.0 | 28.4 |

**Textual Reasoning**   Table 4 details the language modeling performance of MMaDA across a range of benchmarks, encompassing general tasks such as MMLU, ARC-C, and TruthfulQA, as well as mathematical tasks including GSM8K, MATH, and GPQA. Despite being trained on limited task-specific tokens and solely open-source text data, MMaDA achieves comparable performance compared to strong baselines such as Qwen2-7B and LLaMA3-8B on MMLU, ARC-C, and consistently outperforms LLaDA-8B on math benchmarks. Notably, MMaDA (**Ours**) pioneers the joint training of a unified diffusion model for text generation, multimodal reasoning, and image generation—a multi-task configuration rarely explored in prior unified architectures. These results underscore the viability of diffusion-based models as general-purpose LLMs and indicate potential for stronger future performance through enhanced text data and scaling. Qualitative results are in Appendix F.

## 5   Observations, Analysis and Conclusion

**Synergy Across Various Tasks**   Throughout the joint training process, we observe a clear synergy across the three task categories—text generation, multimodal understanding, and image generation. As shown in Fig. 3, all key performance metrics exhibit consistent improvements during Stage 2 (training steps 120K–200K), reflecting the mutually beneficial nature of our unified training framework. This synergy is also evident qualitatively: as illustrated in Fig. 4, the model's responses—both textual and visual—become increasingly complex and coherent. Specifically, textual outputs grow more informative and logically structured, while visual understanding yields more precise and grounded descriptions. Consequently, for the same prompt, the generated images become more accurate, detailed, and better aligned with the given instructions, demonstrating the effectiveness of joint optimization in enhancing cross-modal alignment and compositionality.

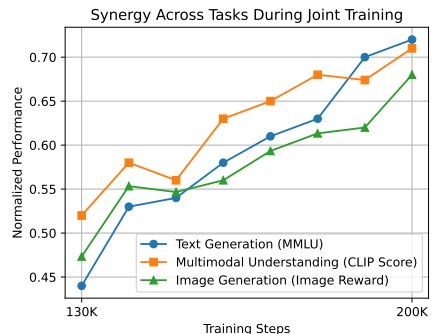

Figure 3: Key Performance Metrics Across Three Tasks.

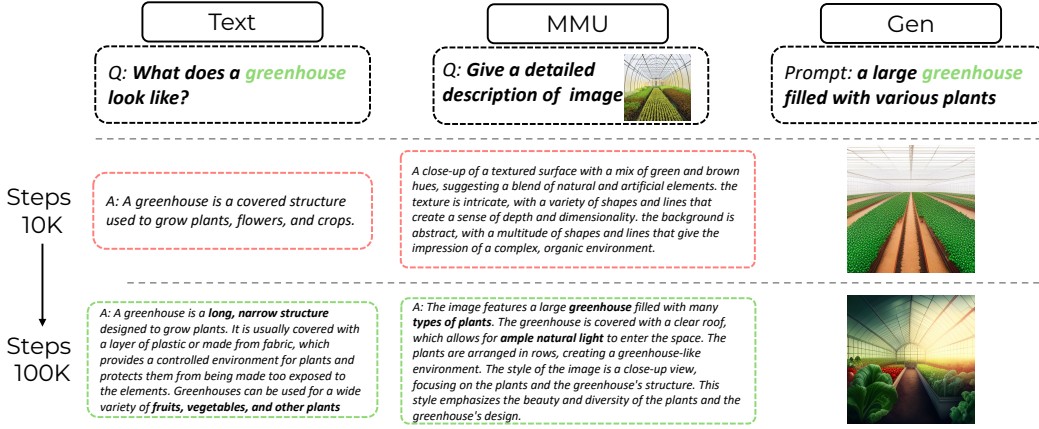

Figure 4: Qualitative Illustration of Synergy Across Modalities.

**Sampling Efficiency** We identify **sampling efficiency** as a key advantage of diffusion models over autoregressive (AR) approaches. Unlike AR models, which generate tokens sequentially, diffusion models enable parallel token generation within each denoising step, substantially reducing the number of forward passes required. To quantify this advantage, we evaluate the performance of MMaDA under varying numbers of denoising steps. In our setup, generation begins from 1024 [MASK] tokens, allowing up to 1024 denoising steps—corresponding to a $512 \times 512$ resolution image. As shown in Table 5, image generation maintains strong performance even with as few as 15 or 50 steps. For text and multimodal tasks, coherent outputs can be achieved with just a quarter or half of the full steps. These results underscore the efficiency potential of diffusion-based language models and suggest that future advances in sampling techniques could further enhance their speed and quality.

Table 5: Generation performance of MMaDA under different denoising steps. **\*** Metrics: CLIP Score for image generation and multimodal understanding, MMLU accuracy for text generation.

| Task | Denoising Steps | Metrics* |
|---|---|---|
| Image Generation | 1024 | 32.8 |
| | 50 | 32.0 |
| | 15 | 31.7 |
| Multimodal Understanding | 1024 | 35.5 |
| | 512 | 36.1 |
| | 256 | 35.4 |
| Text Generation | 1024 | 66.9 |
| | 512 | 66.3 |
| | 256 | 65.7 |

**Task Extension** A notable advantage of diffusion-based models is their natural ability to perform *inpainting* and *extrapolation* without requiring additional fine-tuning. This stems from the fact that these tasks can be formulated as masked token prediction problems, which are inherently integrated into the training objective of diffusion models. While prior work such as Show-o demonstrates this property only in the context of image generation, MMaDA extends it further to multimodal understanding and text generation. As illustrated in Figure 5, our model supports inpainting across three modalities: (i) predicting missing spans in text sequences, (ii) completing answers in visual question answering given an image and partial input, and (iii) performing image inpainting conditioned on incomplete visual prompts. These examples showcase the flexibility and generalization capabilities of our unified diffusion architecture across diverse generation and reasoning tasks.

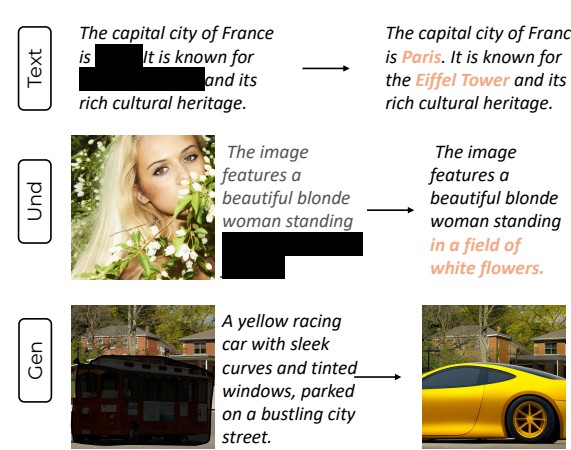

Figure 5: Inpainting Task Extension.

## 6 Conclusion

This work introduces a unified diffusion foundation model, namely MMaDA, that integrates textual reasoning, multimodal understanding, and generation within a single probabilistic framework. To the best of our knowledge, MMaDA is the first to systematically explore the design space of diffusion-based foundation models, proposing novel post-training strategies. Extensive experiments across diverse vision-language tasks demonstrate that MMaDA is comparable to or even better than specialized models, highlighting the potential of diffusion models as a next-generation foundation paradigm for multimodal intelligence. For future work, we aim to improve this new multimodal foundational framework and model from two perspectives: (i) we will try to scale both pre-training and post-training setting for more extensive applications; (ii) we will utilize diffusion-centric reinforcement learning algorithms (e.g., TraceRL [65]) to enhance the capabilities of multimodal large diffusion language models.

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

# A   More Related Works

**Multimodal Large Language Models for Multimodal Understanding**   Recent developments in large language models (LLMs) such as Gemini-2.0 [66], o1-preview [67], and DeepSeek-R1 [68] have improved the evolution of multimodal large language models (MLLMs) [69–71]. Early efforts in this domain, including LLaVA [72], MiniGPT-4 [73], and InstructBLIP [39], showcased impressive capabilities in multimodal understanding. These studies advanced the integration of LLMs into multimodal contexts by projecting features from pre-trained modality-specific encoders, such as CLIP [56], into the input space of LLMs, thereby facilitating multimodal understanding and reasoning within a unified transformer. Many efforts have been made for MLLMs regarding vision encoders, alignment adapters, and curated datasets [74, 75, 40, 76], and most of them follow an autoregressive generation paradigm that has been proven effective for text generation in LLMs. However, they are usually not capable of performing textual and multimodal reasoning concurrently. In this work, MMaDA develops diffusion foundation models to fill this gap.

**Diffusion Models and Autoregressive Models for Visual Generation**   A large number of diffusion models [60, 77, 61, 62, 78–82] have demonstrated notable success in visual generation. In addition to the typical denoising diffusion process on the continuous space, a series of frameworks, such as D3PM [83] and VQ-Diffusion [84], adopt discrete diffusion modeling [83, 85] for visual generation. Specifically, the image is denoted as a sequence of discrete tokens using pretrained image tokenizers [86, 87, 24, 88]. In tthe raining stage, the model is optimized to recover the original values of a portion of these tokens that are randomly masked. Transformer series [89, 90, 1, 2, 91] has demonstrated significant capabilities in autoregressive modeling for NLP tasks. Many approaches [92–96] try to apply the autoregressive modeling to perform visual generation by modeling semantic dependency within visual details. For example, LlamaGen [63] employs Llama architectures [91] and refines codebook design to enhance the performance of discrete tokenizers in class-conditional image generation. VAR [31] replaces the "next-token prediction" paradigm with "next-scale prediction" by designing a multi-scale image tokenizer. However, existing autoregressive methods still lag behind diffusion methods in terms of visual generation capabilities. In this work, MMaDA train diffusion models to model textual and visual contents, and infer efficiently with AR or Semi-AR sampling.

# B   Preliminaries of Discrete Diffusion, PPO and GRPO

## B.1   Discrete Diffusion and Mask Token Prediction

Discrete denoising diffusion models have emerged as a powerful paradigm for modeling discrete data representations, particularly in visual and textual domains. Recent works such as VQ-Diffusion [97], MaskGIT [98], and Muse [99] demonstrate their effectiveness in generating high-quality discrete tokens. Building on these advancements, Show-o [20] unifies discrete diffusion and mask token prediction into a single framework, enabling joint optimization of noise corruption and semantic recovery. Below, we formalize the key components of this discrete diffusion process.

### B.1.1   Forward and Reverse Diffusion Processes

The *forward* diffusion process progressively corrupts the initial data $x_0$ (e.g., a sequence of visual/textual tokens) into noisy latent variables $x_1, ..., x_T$ via a fixed Markov chain $q(x_t|x_{t-1})$. At each step, this process stochastically perturbs tokens—such as replacing them with uniform noise or introducing a special [MASK] token. The *reverse* process, parameterized by a learned model $p_\theta$, reconstructs $x_0$ from $x_T$ by sequentially sampling $q(x_{t-1}|x_t, x_0)$.

To formalize this, consider a single token $x_0^i$ at position $i$ in $x_0$, where $x_0^i \in \{1, 2, ..., K\}$ corresponds to an entry in a codebook. For simplicity, we omit the index $i$ in subsequent derivations. The transition probabilities between consecutive tokens are defined by a matrix $Q_t \in \mathbb{R}^{K \times K}$, with $[Q_t]_{mn} = q(x_t = m|x_{t-1} = n)$. The forward Markov process for the entire token sequence is then expressed as:

$$q(x_t|x_{t-1}) = v^\top(x_t)Q_t v(x_{t-1}), \tag{7}$$

where $v(x)$ is a one-hot vector of length $K$ (with 1 at position $x$). The categorical distribution over $x_t$ is derived from $Q_t v(x_{t-1})$.

A critical property of the Markov chain allows us to marginalize intermediate steps and compute the direct transition from $x_0$ to $x_t$:

$$q(x_t|x_0) = \boldsymbol{v}^\top(x_t)\overline{\boldsymbol{Q}}_t\boldsymbol{v}(x_0), \quad \text{with } \overline{\boldsymbol{Q}}_t = \boldsymbol{Q}_T\boldsymbol{Q}_{T-1}\cdots\boldsymbol{Q}_1. \tag{8}$$

Furthermore, the posterior distribution conditioned on $x_0$ is analytically tractable:

$$q(x_{t-1}|x_t, x_0) = \frac{q(x_t|x_{t-1}, x_0)q(x_{t-1}|x_0)}{q(x_t|x_0)} = \frac{\left(\boldsymbol{v}^\top(x_t)\boldsymbol{Q}_t\boldsymbol{v}(x_{t-1})\right)\left(\boldsymbol{v}^\top(x_{t-1})\overline{\boldsymbol{Q}}_{t-1}\boldsymbol{v}(x_0)\right)}{\boldsymbol{v}^\top(x_t)\overline{\boldsymbol{Q}}_t\boldsymbol{v}(x_0)}. \tag{9}$$

This property is essential for deriving the variational lower bound of the diffusion process.

### B.1.2 Transition Matrix Design and Masking Strategy

The design of the transition matrix $\boldsymbol{Q}_t$ is pivotal to the success of discrete diffusion models. Early works [100, 101] propose injecting uniform noise into the categorical distribution, leading to:

$$\boldsymbol{Q}_t = \begin{bmatrix} \alpha_t + \beta_t & \beta_t & \cdots & \beta_t \\ \beta_t & \alpha_t + \beta_t & \cdots & \beta_t \\ \vdots & \vdots & \ddots & \vdots \\ \beta_t & \beta_t & \cdots & \alpha_t + \beta_t \end{bmatrix}, \tag{10}$$

where $\alpha_t \in [0, 1]$ controls the retention probability, and $\beta_t = (1 - \alpha_t)/K$ ensures uniform diffusion across all $K$ categories. However, this approach often introduces abrupt semantic changes due to aggressive replacement of tokens with unrelated categories.

To address this limitation, Show-o adopts a **mask-and-replace** strategy inspired by masked language modeling. A special [MASK] token is introduced, expanding the token space to $K + 1$ states. The transition matrix is redefined as:

$$\boldsymbol{Q}_t = \begin{bmatrix} \alpha_t + \beta_t & \beta_t & \cdots & \beta_t & 0 \\ \beta_t & \alpha_t + \beta_t & \cdots & \beta_t & 0 \\ \vdots & \vdots & \ddots & \vdots & \vdots \\ \beta_t & \beta_t & \cdots & \alpha_t + \beta_t & 0 \\ \gamma_t & \gamma_t & \cdots & \gamma_t & 1 \end{bmatrix}, \tag{11}$$

where: - Ordinary tokens have a probability $\alpha_t$ to remain unchanged, $\beta_t$ to be uniformly diffused, and $\gamma_t$ to be replaced by [MASK]. - The [MASK] token retains its state with probability 1.

This design explicitly signals corrupted positions during the forward process, enabling the reverse network to focus on reconstructing masked regions. The parameters satisfy $\alpha_t + K\beta_t + \gamma_t = 1$, ensuring proper normalization.

### B.1.3 Variational Objective and Loss Simplification

Following the D3PM framework [101], the variational lower bound for the discrete diffusion process is derived as:

$$\mathbb{E}_{q(\mathbf{x}_0)}[\log p_\theta(\mathbf{x}_0)] \geq -\mathcal{L}_{\text{ELBO}}(\mathbf{x}_0, \theta) \geq \sum_{t=1}^{T}\mathbb{E}_{q(\mathbf{x}_0, \mathbf{x}_t)}[\log p_\theta(\mathbf{x}_0|\mathbf{x}_t)] + C. \tag{12}$$

To simplify training, Show-o leverages the mask token prediction objective. Specifically, the model learns a neural network $p_\theta$ to reconstruct masked tokens in $\mathbf{x}_0$ from the noised $\mathbf{x}_t$, following the methodology of MaskGIT [98]. This approach avoids explicit modeling of the full posterior and instead focuses on recovering only the corrupted regions, significantly reducing computational complexity.

### B.2 Proximal Policy Optimization and Group Relative Policy Optimization

Proximal Policy Optimization (PPO) [102] is a widely adopted reinforcement learning algorithm that balances policy updates with stability through a clipped surrogate objective. The core idea of PPO lies

in constraining policy changes to a proximal region around the previous policy, thereby preventing large, destabilizing updates. This is achieved by introducing a clipping mechanism that bounds the importance sampling ratio during optimization. Specifically, PPO maximizes the following objective:

$$\mathcal{J}_{\text{PPO}}(\theta) = \mathbb{E}_{(q,a)\sim\mathcal{D}, o_{\leq t}\sim\pi_{\theta_{\text{old}}}(\cdot|q)}$$
$$\left[ \min\left( \frac{\pi_\theta(o_t \mid q, o_{<t})}{\pi_{\theta_{\text{old}}}(o_t \mid q, o_{<t})}\hat{A}_t, \ \text{clip}\left( \frac{\pi_\theta(o_t \mid q, o_{<t})}{\pi_{\theta_{\text{old}}}(o_t \mid q, o_{<t})}, 1-\varepsilon, 1+\varepsilon \right)\hat{A}_t \right) \right], \quad (13)$$

where $(q, a)$ denotes a question-answer pair sampled from the dataset $\mathcal{D}$, $\varepsilon$ is the clipping threshold, and $\hat{A}_t$ is the estimated advantage at time step $t$. The advantage $\hat{A}_t$ is typically computed using Generalized Advantage Estimation (GAE) [103], which combines multiple temporal differences with discounting and mixing parameters $(\gamma, \lambda)$:

$$\hat{A}_t^{\text{GAE}(\gamma,\lambda)} = \sum_{l=0}^{\infty} (\gamma\lambda)^l \delta_{t+l}, \quad \text{with } \delta_l = R_l + \gamma V(s_{l+1}) - V(s_l). \quad (14)$$

This formulation ensures robustness against high-variance estimates while maintaining compatibility with off-policy data.

In contrast, Group Relative Policy Optimization (GRPO) [33] introduces two key innovations to address limitations in PPO's value function dependency and global advantage estimation. First, GRPO eliminates the explicit modeling of the value function $V(s)$ and instead computes advantages in a group-relative manner. For each $(q, a)$ pair, the behavior policy $\pi_{\theta_{\text{old}}}$ generates $G$ responses $\{o_i\}_{i=1}^G$, and the advantage of each response is normalized relative to its group:

$$\hat{A}_{i,t} = \frac{r_i - \text{mean}(\{R_i\}_{i=1}^G)}{\text{std}(\{R_i\}_{i=1}^G)}, \quad (15)$$

where $r_i$ represents the raw reward for response $o_i$. This design enables GRPO to focus on relative performance within a local context, reducing sensitivity to absolute reward scales.

Second, GRPO extends PPO's clipped objective by incorporating an explicit KL divergence penalty term between the current policy $\pi_\theta$ and a reference policy $\pi_{\text{ref}}$. The final objective is defined as:

$$\mathcal{J}_{\text{GRPO}}(\theta) = \mathbb{E}_{(q,a)\sim\mathcal{D}, \{o_i\}_{i=1}^G \sim \pi_{\theta_{\text{old}}}(\cdot|q)}$$
$$\left[ \frac{1}{G}\sum_{i=1}^G \frac{1}{|o_i|}\sum_{t=1}^{|o_i|} \left( \min\left( r_{i,t}(\theta)\hat{A}_{i,t}, \ \text{clip}\left( r_{i,t}(\theta), 1-\varepsilon, 1+\varepsilon \right)\hat{A}_{i,t} \right) - \beta D_{\text{KL}}(\pi_\theta\|\pi_{\text{ref}}) \right) \right], \quad (16)$$

where $r_{i,t}(\theta) = \frac{\pi_\theta(o_{i,t}|q,o_{i,<t})}{\pi_{\theta_{\text{old}}}(o_{i,t}|q,o_{i,<t})}$ is the importance sampling ratio at time $t$. Notably, GRPO computes the loss at the sample level: for each sequence, the per-token loss is averaged first, followed by averaging across all sequences in the group. This hierarchical aggregation enhances stability in multi-response settings.

## C  Sampling Details

**Text Generation**  For text generation, we adopt the semi-autoregressive denoising strategy introduced in LLaDA [22], which integrates autoregressive decoding with diffusion-based denoising. Specifically, the output sequence is partitioned into multiple blocks and generated from left to right. Within each block, logits are computed for all masked positions, and a subset of tokens is selected—either randomly or based on confidence scores—for denoising. The masking schedule follows a linear schedule, consistent with LLaDA. The denoising process is repeated for given steps.

In our evaluation, we set the total sequence length to $N = 1024$ and perform $\frac{N}{2} = 512$ denoising steps. The sequence is divided into blocks of 64 tokens. At each step, we unmask the 2 tokens with the lowest confidence within the current block, irrespective of their positions. Once all tokens in a block are denoised, the process proceeds to the next block. A qualitative comparison is provided below. As shown, a semi-autoregressive denoising strategy tends to generate more intricate and detailed descriptions, whereas non-autoregressive fixed-length generation often produces very short responses.

This observation is consistent with findings reported for LLaDA. For instruction-tuned models, given that the training process incorporates a substantial number of |EOS| tokens, directly applying the lowest-confidence remasking strategy without dividing into blocks leads to an unnaturally high frequency of |EOS| tokens in the generated sentences.

---

**Qualitative Comparison of Different Sampling strategy**

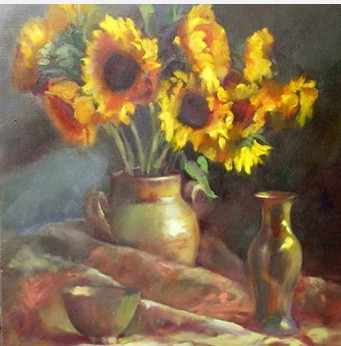

**Prompt:** Please describe this image in detail.

---

**Fix-length Generation ( Length = 1024, Steps = 512, Block Size = 1024)**
The image is a detailed oil painting depicting a still life scene.

---

**Semi-Autoregressive ( Length = 1024, Steps = 512, Block Size = 64)**
The painting depicts a serene still life scene on a tablecloth with a textured, multicolored pattern. In the center, there is a rustic, cream-colored ceramic pitcher with brown accents, holding a vibrant bouquet of sunflowers. The sunflowers, with their bright yellow petals and deep brown centers, are arranged in a way that they seem to burst forth from the pitcher, creating a sense of abundance and vitality. To the right of the pitcher, there is a small, ornate glass vase with a patina finish, adding an element of contrast and visual interest to the composition. In the foreground, a sliced loaf of bread is placed on the tablecloth, with a piece of bread and a knife nearby, suggesting a meal or snack. The background is softly blurred, focusing attention on the main elements of the painting. The overall composition is balanced and harmonious, with the warm tones of the sunflowers and ceramic pitcher complementing the cooler bronze of the vase and the soft pastels of the bread.

---

**Image Generation** For image generation, we adopt a low-confidence remasking strategy and follow a cosine noise schedule, consistent with the setup in MagViT v2. In contrast to text generation, we do not employ a semi-autoregressive approach; instead, the entire output sequence is treated as a single generation block. During evaluation, we generate sequences of length 1024, corresponding to $512\times512$ resolution images. The denoising process consists of 50 timesteps, and we apply classifier-free guidance with a guidance scale set to 3.5.

## D    Details of UniGRPO

In this section, we detail our UniGRPO training process and illustrate the differences compared to LLaDA and d1 (diff-GRPO) [34]. Notably, LLaDA does not incorporate a reinforcement learning procedure, yet it includes an algorithm for computing log probabilities. Diff-GRPO introduced the first GRPO-style RL algorithm for diffusion language models and proposed an alternative method to compute log probabilities for a given question-answer pair. We first outline the methodologies of these two prior works:

**(1) LLaDA** While LLaDA does not employ an RL process, it offers a method to estimate the log probability of a given pair $(q, a)$ using Monte Carlo simulation. For a given answer, N mask ratios are randomly sampled from the interval $(0, 1)$ (where $N = 128$ in the original work). Subsequently, a batch of $N$ inputs is constructed. Each input instance consists of the original question tokens concatenated with the answer, where a portion of the answer tokens (corresponding to the sampled mask ratio) is replaced by [MASK] tokens. A model forward pass is performed on this batch, and the logits corresponding to the masked regions are collected and averaged to obtain the log probability

---

**Algorithm 1** UniGRPO Policy Gradient Optimization

---

**Require:** Reference model $\pi_{\text{ref}}$, prompt distribution $\mathcal{D}$, number of completions per prompt $G$, number of inner updates $\mu$, diffusion steps $T$
1: Initialize policy $\pi_\theta \leftarrow \pi_{\text{ref}}$
2: **while** not converged **do**
3:     $\pi_{\text{old}} \leftarrow \pi_\theta$
4:     Sample a prompt $q \sim \mathcal{D}$
5:     Sample $G$ completions $o_i \sim \pi_{\text{old}}(\cdot \mid q)$, for $i \in [G]$
6:     For each $o_i$, compute reward $r_i$ and advantage $A_i^k(\pi_{\text{old}})$ using Eq. (15)
7:     Sample a starting timestep $t_0 \sim \mathcal{U}(0, T-1)$
8:     Generate $\mu - 1$ uniformly spaced timesteps $t_1, \ldots, t_{\mu-1}$ from $[t_0, T]$
9:     **for** gradient update iterations $n = 1, \ldots, \mu$ **do**
10:         **if** $n = 1$ **then**
11:             Sample a starting mask ratio $r_1 \sim \mathcal{U}(0, 1)$ and compute initial timestep $t_1 = \lfloor r_1 \cdot T \rfloor$
12:         **else**
13:             Uniformly divide remaining timesteps: $t_n = \lfloor \frac{(n-1)}{(\mu-1)} \cdot (T - t_1) + t_1 \rfloor$
14:         Construct input $(q, \text{masked } o_i)$ using timestep $t_n$ (with $q$ always unmasked)
15:         For $\pi_\theta, \pi_{\text{old}}, \pi_{\text{ref}}$, estimate log-probabilities of masked tokens in $o_i$ at $t_n$
16:         Compute UniGRPO objective Eq. (5) and update $\pi_\theta$ via gradient descent
17: **return** $\pi_\theta$

---

$\log p(q, a)$. The principal drawback of this method is its significant computational overhead, rendering it impractical for on-policy RL algorithms.

**(2) d1 (diff-GRPO)** The d1 framework [34] introduced the first GRPO-style RL algorithm for diffusion language models. To overcome the efficiency issues of LLaDA, d1 employs a novel masking strategy. In each iteration, only a single forward pass is performed, eschewing Monte Carlo simulation. The input is constructed by applying a random mask to the question tokens and completely masking all answer tokens. The stochasticity is intended to be achieved by selecting different tokens for masking across different iterations, even with the same mask ratio. While this approach in d1 enables GRPO training, we identify potential limitations:

- **Question Masking:** The random masking of question tokens is of unclear practical significance. In typical question-answering scenarios, the question is always fully observed during both training and inference. We argue that such random masking of questions serves primarily to introduce stochasticity without direct relevance to the task's practical application.

- **Answer Masking Strategy:** By consistently masking the entire answer, the model is effectively trained only on the initial denoising step (i.e., predicting the full sequence from a fully masked state). We contend that this approach provides insufficient learning depth, potentially causing the model to behave more like a single-step predictive AR model rather than leveraging the multi-step denoising capabilities inherent to diffusion models. This underutilizes the diffusion model's unique strengths.

As introduced in Section 3.3.1, our UniGRPO addresses the masking issues with the following key modifications:

1. **Unmasked Questions:** In UniGRPO, all question tokens remain unmasked during training. This aligns the training process with the inference conditions and practical use-cases where the question is always fully provided.

2. **Iteratively Varied Answer Masking:** We apply a random mask ratio to the answer tokens. Crucially, this mask ratio, denoted $\rho_a$, varies uniformly with the training iteration $\mu$ (e.g., $\rho_a \sim U(0, 1)$ sampled anew or cycled through discrete steps each iteration). This strategy aims to preserve stochasticity while ensuring the model is exposed to various stages of the diffusion denoising process, from nearly fully masked to nearly fully denoised answers. By doing so, UniGRPO learns from multi-step denoising information, which is consistent with

conventional training methodologies for diffusion models and allows for the full utilization of their multi-step generative power.

Through this design, UniGRPO captures the essential multi-step denoising dynamics of diffusion models. By allowing the model to predict answers under diverse masking conditions while preserving the natural structure of the input, it avoids the pitfalls of both computational inefficiency (as in LLaDA) and oversimplified prediction (as in d1). The training procedure of UniGRPO is outlined in Algorithm 1. At each iteration, we sample a mask ratio $r_\mu$ uniformly from a predefined range, apply this ratio to the answer tokens, and perform a single forward pass using the unmasked question and masked answer. We then compute token-level log-likelihoods in the masked regions, apply the GRPO objective, and update the policy accordingly.

## E   Ablation Studies

### E.1   General Ablations on Mixed Long-CoT Fine-tuning and UniGRPO

We present quantitative ablation results of our MMADA across different training stages: Mixed Long-CoT fine-tuning and UniGRPO. All results generated follow the sampling process in Appendix C. As shown in the Table 6, after Stage 1, our model still lags behind most baselines. In Stage 2, Mixed Long-CoT fine-tuning substantially enhances the model's reasoning capabilities, particularly in mathematical and geometric domains. In Stage 3, UniGRPO further improves performance, allowing the model to achieve results comparable to state-of-the-art methods across various tasks, including mathematical reasoning, geometric problem-solving, and image generation benchmarks such as CLIP Score and ImageReward. These results demonstrate that UniGRPO effectively boosts both the model's understanding/reasoning and generative capabilities.

Table 6: Ablations on Mixed Long-CoT fine-tuning and UniGRPO

| Model | GSM8K | MATH500 | GeoQA | CLEVR | CLIP Score | ImageReward |
|---|---|---|---|---|---|---|
| MMADA After Stage 1 | 17.4 | 4.2 | 8.3 | 10.3 | 23.1 | 0.69 |
| + Mixed Long-CoT Finetuning | 65.2 | 26.5 | 15.9 | 27.5 | 29.4 | 0.84 |
| + UniGRPO (MMADA) | 73.4 | 36.0 | 21.0 | 34.5 | 32.5 | 1.15 |

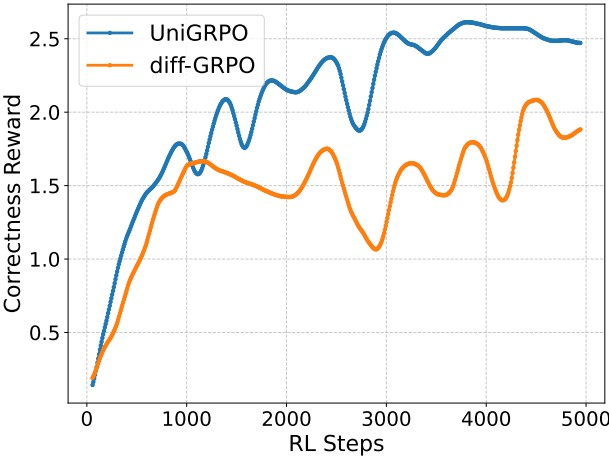

Figure 6: Comparison of different masking strategies on GSM8K reward trends during training.

### E.2   Design Choices of UniGRPO

**Effect of General Masking Strategy**   To evaluate the impact of our proposed masking strategy, we first conduct a comparative analysis with d1 [34] within our reinforcement learning framework. Given the substantial computational cost associated with large-scale ablation studies, we perform

these experiments on the GSM8K dataset, utilizing 8 A100 GPUs. Both the original d1 methodology and our UniGRPO approach are applied to this dataset, starting from the same pre-trained checkpoint of our MMaDA. We present the reward trends during training in Fig. 6. As shown in the figure, our method consistently achieves higher reward values during training, aligning well with our theoretical analysis. In contrast to d1, UniGRPO removes masking from the question and applies partial masking to the answer rather than masking it entirely. This results in input sequences that retain partial noise, encouraging the model to learn across multiple denoising timesteps. Consequently, this better leverages the intrinsic characteristics of diffusion models and improves the overall learning capacity.

**Effect of Uniformly Random masking**   In place of fully random masking across iterations, we adopt a uniformly random masking strategy for the answer portion. Specifically, we first sample a random starting timestep, and then uniformly generate the remaining denoising timesteps across the full diffusion timesteps (set to 1000 in our experiments). For instance, given a randomly selected starting timestep of 100 and a total of 5 training iterations, the remaining timesteps are uniformly spaced and set to 300, 500, 700, and 900. This design ensures a more consistent coverage of the diffusion process while retaining randomness.

We illustrate the training reward trends resulting from this structured masking strategy in Fig. 7. As shown, the baseline approach with fully random timestep selection tends to introduce instability during training, leading to more frequent reward fluctuations and requiring a greater number of steps to converge. In contrast, our uniformly spaced sampling strategy effectively approximates the behavior of Monte Carlo averaging in log-likelihood estimation, resulting in improved stability and faster convergence.

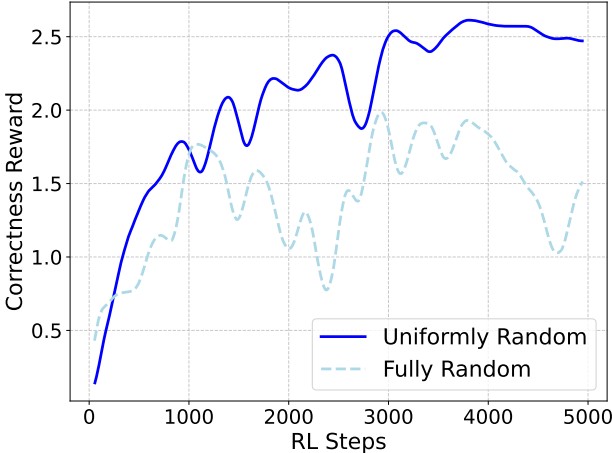

Figure 7: Comparison of different random masking strategies on GSM8K reward trends during training.

# F  Qualitative Comparisons (with Reasoning CoT)

## Qualitative Comparison of World Knowledge-Aware Text-to-Image Generation (1)

**Prompt:** The largest terrestrial carnivore from the Arctic.

**Other models:**

| Show-o | Emu3 | Janus Pro 7B |
|---|---|---|
| 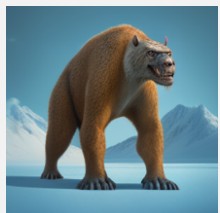 | 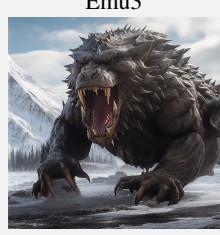 | 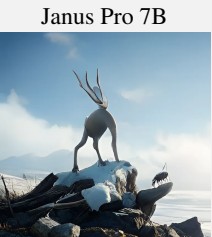 |

**MMaDA:**

`<think>`
`largest terrestrial carnivore from the Arctic is the Polar Bear.`
`</think>`
`A majestic polar bear stands on a snowy landscape, its fur glowing against the icy backdrop.`

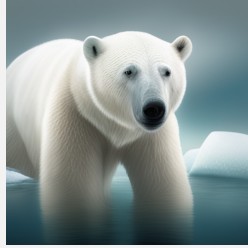

## Qualitative Comparison of World Knowledge-Aware Text-to-Image Generation (2)

**Prompt:** A famous landmark building in Europe.

**Other models:**

| Show-o | Emu3 | Janus Pro 7B |
|---|---|---|
| 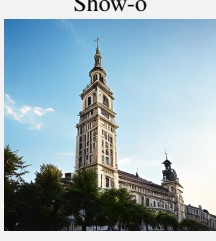 | 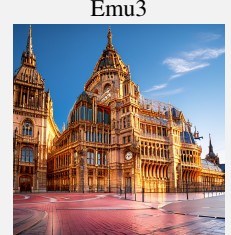 | 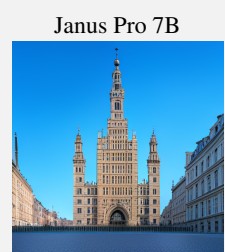 |

**MMaDA:**

`<think>`
`A famous landmark building in Europe is likely the Eiffel Tower.`
`</think>`
`The Eiffel Tower is hollow, with a cloudy and muted sky.`

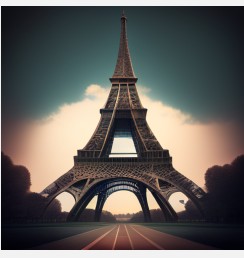

Qualitative Comparison of World Knowledge-Aware Text-to-Image Generation (3)

**Prompt:** A traditional strong spirit from Russia.

**Other models:**

| Show-o | Emu3 | Janus Pro 7B |
|:------:|:----:|:------------:|
| 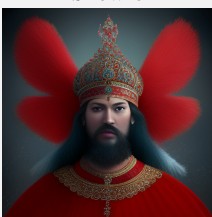 | 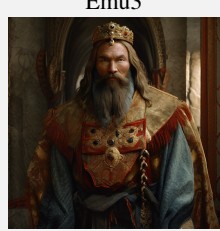 | 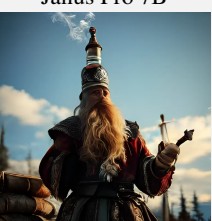 |

**MMaDA:**

```
<think>
The user is describing a traditional strong spirit from Russia.  A well-known spirit that fits
this description is vodka.
</think>
An silver bottle of vodka sits next to a decanter and a halved lime.
```

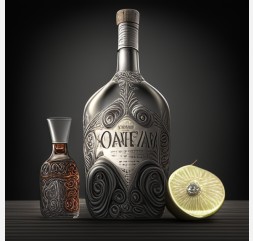

Qualitative Comparison of World Knowledge-Aware Text-to-Image Generation (4)

**Prompt:** A famous statue gifted to the United States by France.

**Other models:**

| Show-o | Emu3 | Janus Pro 7B |
|:------:|:----:|:------------:|
| 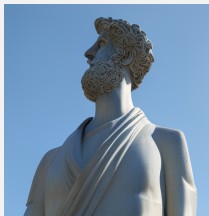 | 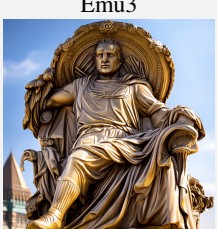 | 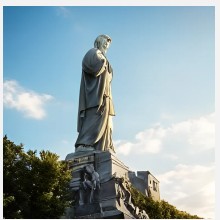 |

**MMaDA:**

```
<think>
The description refers to a well-known landmark that was given from France to the United States.
This is known as the Statue of Liberty.
</think>
A low-angle shot of the Statue of Liberty against a blue sky with a few wispy clouds.
```

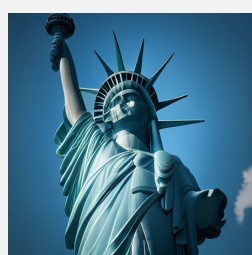

**Question:** How many plums will balance one pear, given that 3 apples and one pear weigh as much as 10 plums, and one apple and 6 plums balance one pear? Assume that fruits of the same kind have the same weight.

---

**Show-o:**

*< Show-o does not support this task >*

---

**Emu3:**

*< Emu3 does not support this task >*

---

**Janus Pro 7B:**

*< Janus Pro 7B does not support this task >*

---

**MMaDA:**

`<think>`
Okay, let's see. I need to figure out how many plums are needed to balance the weight of a single pear. The problem gives me some information. Let me try to parse it step by step.
First, the problem says: 3 apples together with a pear have the same weight as 10 plums. Then, the combination of one apple and 6 plums weighs the same as a pear. I need to find out how many plums are needed to equal the weight of a single pear. Since all fruits are equal in weight, probably the same.
Let me start by writing this down. Let's assign variables. Let's let $A$ represent the weight of an apple, $R$ as the weight of a pear, and $M$ as the weight of a plum. So, based on the first statement given:
3A + R = 10M
Right, because 3 apples plus a pear equals 10 plums.
Second, saying one apple and 6 plums weigh the same as a pear. So, A + 6M = R
That's because all apples and plums are equal. So,
A = R - 6M
So, now we have two equations:
1) 3A + R = 10M
2) A + 6M = R
Our goal is to find out how many plums are needed to balance one pear. That is, we need to find the value of $n$ in $R = nM$.
From equation (2): $A + 6M = R \implies A = R - 6M$.
Now substitute this value of $A$ into equation (1):
3A + R = 10M 3(R - 6M) + R = 10M 3R - 18M + R = 10M 4R - 18M = 10M 4R = 28M R = 7M
So one pear weighs as much as 7 plums.
Let's check with equation (2): A + 6M = R A = R - 6M But if $R = 7M$, then $A = 7M - 6M = M$.
So an apple weighs as much as 1 plum, which works when we check the first equation: 3A + R = 10M
3(M) + 7M = 10M 3M + 7M = 10M
So both equations are satisfied.
Therefore, the answer is 7 plums.
`</think>`

To determine how many plums are to balance a single pear, we set up the equations based on the given conditions:
1. Let $A$ represent the weight of an apple, $R$ represent the weight of a pear, and $M$ the weight of a plum.
2. The equations are:
- Equation 1: $3A + R = 10M$
- Equation 2: $A + 6M = R$ (1 apple + 6 plums equals a pear)
**Step 1: Solve for $A$ in terms of $R$ and $M$**
From Equation 2: $A = R - 6M$
**Step 2: Substitute $A$ into Equation 1**
$3(R - 6M) + R = 10M$
$3R - 18M + R = 10M$
$4R - 18M = 10M$
$4R = 28M$
$R = 7M$
**Final Answer:**
$\boxed{7}$ plums are needed to balance the weight of a single pear.

**Question:** How many positive divisors does the integer $n$ possess, where $n = 7^3 \times 11^2 \times 13^4$?

---

**Show-o:**
< *Show-o does not support this task* >

---

**Emu3:**
< *Emu3 does not support this task* >

---

**Janus Pro 7B:**
< *Janus Pro 7B does not support this task* >

---

**MMaDA:**
<think>
Okay, so I need to figure out how many positive divisors the integer n has, where n is given as 7 cubed times 11 squared times 13 to the fourth power. Hmm, let me think. I remember that the number of divisors of a number is related to its prime factorization exponents. Let me try to recall the general formula here.

First, right that if a number is expressed in a product of primes, like a number in the form $n = p_1^a \times p_2^b \times \ldots \times p_k^c$, then the number of positive divisors is (a + 1)(b + 1)...(c + 1). Right? Because for each exponent in the prime factorization, you add 1, then multiply them all together for the total number of divisors.

Let me verify that. Take an example: if n is $2^1 \times 3^1$, then the positive divisors are 1, 2, 3, 6. Using the formula, (1+1)*(1+1) = 2*2 = 4, which checks out. Another example: like $2^2 \times 3^1$. Then the divisors are 1, 2, 4, 3, 6, 12. That makes 6 divisors. The formula gives (2+1)*(1+1) = 3*2 = 6, so that's correct. Another example: $2^2 \times 5^2$. Then divisors count is (2+1)*(2+1) = 3*3=9. So the formula seems correct.

So, applying this formula to n in the given problem: n is $7^3 \times 11^2 \times 13^4$. Here, the exponents are 3, 2, and 4. So, applying the number of divisors formula should be (3 + 1)(2 + 1)(4 + 1). Let's compute that step by step.

First, calculate each exponent plus one: 3 + 1 is 4. 2 + 1 is 3. 4 + 1 is 5. Then, multiplying those results together: 4 * 3 * 5. Let me check that. 4 times 3 is 12, then 12 times 5 is 60. So the number of positive divisors is 60.

Wait, but let me make sure I didn't make a mistake in the multiplication. Let me verify the steps again. 4*3 is 12, 12*5 is 60. Yes, that seems correct.

But just to be thorough, let me write the original number n as $7^3 \times 11^2 \times 13^4$. The prime factors are 7, 11, 13. Each raised to the exponents 3, 2, and 4, respectively. Since they are all prime, the formula for divisors uses the exponents as factors plus 1. Then multiplying the results: 4*3*5=60.

Is there any possible way this could be wrong? Well, maybe if the exponents were different, but the original expression is already in its exponential form. Let me check if I misread. For example, if one of the exponents was not what is stated, but the problem writes $7^3$ times $11^2$ times $13^4$, so the exponents are 3, 2, 4, that's correct.

Alternatively, could there be a mistake in the multiplication? Let me check 4*3*5 again. 4*3 is 12, 12*5 is 60. Correct. So 60 is the correct answer.

Wait, just to be thorough, let's list the combinations: any divisor is $7^a \times 11^b \times 13^c$, where $0 \leq a \leq 3$, $0 \leq b \leq 2$, and $0 \leq c \leq 4$. The number of choices for a is 4, for b is 3, for c is 5, so 4*3*5 = 60 possible positive divisors. Perfect.

I don't see any mistakes here. The answer is 60.
</think>

**Solution:**
The integer $n$ is given as $n = 7^3 \times 11^2 \times 13^4$. To determine the number of positive divisors of $n$, we use the formula for the number of divisors: if a number is in the form $p_1^a \times p_2^b \times \cdots \times p_k^c$, then it has $(a+1)(b+1)\ldots(c+1)$ divisors.

**Step-by-Step Solution:**
1. **Identify the Exponent:**
- $7^3$: Exponent = 3
- $11^2$: Exponent = 2
- $13^4$: Exponent = 4

2. **Apply the Divisor Formula:**
- For each prime exponent, $(a+1)$, and then multiply:
- For exponent 3: $3 + 1 = 4$
- For exponent 2: $2 + 1 = 3$
- For exponent 4: $4 + 1 = 5$
- Multiply the results: $4 \times 3 \times 5 = 60$

**Answer:** The integer $n$ has $\boxed{60}$ positive divisors.

## Qualitative Comparison of Textual Reasoning (3)

**Question:** A "Hope Number" is a natural number with an odd number of positive divisors. Find the largest "Hope Number" within 1000.

**Show-o:**

*< Show-o does not support this task >*

**Emu3:**

*< Emu3 does not support this task >*

**Janus Pro 7B:**

*< Janus Pro 7B does not support this task >*

**MMaDA:**

<think>
Okay, so I need to find the largest "Hope Number" which is a natural number with an odd number of positive divisors. Let me start by recalling what a Hope Number is. The question says a number has an odd number of divisors. Hmm, that means a number has an odd number of different divisors.
First, I remember that numbers have divisors related to their prime factorization. If a number has the prime factorization $n = p_1^{a_1} p_2^{a_2} \ldots p_k^{a_k}$, the number of divisors is $(a_1 + 1)(a_2 + 1) \cdots (a_k + 1)$. This product is odd if and only if every $a_i + 1$ is odd, i.e., if every exponent $a_i$ is even. That means the number itself is a perfect square (since each exponent is even).
To check: for example, take 9. Its prime factorization is $3^2$. Its divisors are 1, 3, 9. That's three divisors, which is odd. Another example, 16. Prime factorization is $2^4$. Its divisors are 1, 2, 4, 8, 16. That's five divisors, which is odd. How about 15? Its prime factorization is $3 \times 5$, which is not a perfect square. Its divisors are 1, 3, 5, 15-that's four divisors, which is even. Therefore, only perfect squares have an odd number of divisors.
Therefore, all Hope Numbers less than 1000 are perfect squares.
Therefore, the problem reduces to finding the largest perfect square less than or equal to 1000. Let's compute that.
The square root of 1000 is approximately 31.62. So we need to check the largest integer $n$ such that $n^2 \leq 1000$. Let's check 31.
$31^2 = 961$. $32^2 = 1024$, which is greater than 1000.
So $31^2 = 961$ is the largest perfect square within 1000. Therefore, 961 is the largest Hope Number within 1000.
But wait, let me confirm that 961 is indeed a Hope Number. Let's count its divisors. Its prime factorization is $31^2$, and 31 is prime. The exponents are all even, so the number of divisors is $2 + 1 = 3$: 1, 31, 961, which is odd. That checks out.
And to be thorough, is there a perfect square between 961 and 1000? $32^2 = 1024 > 1000$, so there are none.
Therefore, the largest Hope Number within 1000 is 961.
</think>

The largest "Hope Number" within 1000 is **961**.

**Step-by-Step Explanation:**

1. **Understanding Hope Property:** A Hope Number has an odd number of positive divisors. A number has an odd number of divisors **if and only if it is a perfect square** (because in the divisor formula, all exponents must be even so all $a_i + 1$ are odd, and their product is odd).

2. **Largest Hope Number:** Compute perfect squares $\leq 1000$. $31^2 = 961$ and $32^2 = 1024$. Since $32^2 > 1000$, the largest one is $31^2 = 961$.

3. **Verification:** Any non-square number below 1000 has an even number of divisors; only perfect squares have an odd number of divisors.

**Answer:** 961

**Question:** What is the result of $\frac{(20-16)\times(12+8)}{4}$ ?

---

**Show-o:**
*< Show-o does not support this task >*

---

**Emu3:**
*< Emu3 does not support this task >*

---

**Janus Pro 7B:**
*< Janus Pro 7B does not support this task >*

---

**MMaDA:**
<think>
Okay, let me try to figure out the result of this expression: $\frac{(20-16)\times(12+8)}{4}$. Hmm, let's break this down step by step. First, I need to make sure I apply the order of operations correctly. Right, order of operations: parentheses first, then multiplication and division from left to right, then addition and subtraction from left to right. Let me write the expression again to visualize it better: $\frac{(20-16)\times(12+8)}{4}$.

So, the parentheses are within the numerator, which is part of the fraction, and the denominator is 4. According to PEMDAS, I should handle the parentheses first before doing the multiplication and division. Let me look at the parentheses part: (20 - 16) and (12 + 8). Both of those are inside parentheses, so I should compute each part separately.

Starting with the first parentheses: 20 - 16. That's straightforward. 20 minus 16 equals 4. Okay, so that part simplifies to 4.

Next, the second part inside the parentheses: 12 + 8. That's addition, so 12 plus 8 is 20. So now, the numerator becomes $4 \times 20$.

So now, the numerator is 4 × 20. Let me do that multiplication. 4 times 20 is 80. So the numerator simplifies to 80.

Now the entire expression is $\frac{80}{4}$. Dividing that number by 4. Let me check that division. 4 goes into 80 twenty times (since 4 × 20 is 80), so 80 ÷ 4 = 20. So the result is 20.

Wait a second, let me verify that again to make sure I didn't make a mistake. Let's go through each step again.

Original expression: $\frac{(20-16)\times(12+8)}{4}$.
First parentheses: 20 - 16 = 4. Check.
Second parentheses: 12 + 8 = 20. Check.
Multiply those two: 4 × 20 = 80. Check.
Divide by 4: 80 ÷ 4 = 20. Yep, that seems right.

But wait, maybe there's another way to approach this. Sometimes with order of operations, parentheses are easier to catch. Let's see. The original expression is $\frac{(20-16)\times(12+8)}{4}$. The parentheses in the numerator give us multiplication, and the denominator is 4. You could write it as $\frac{4\times20}{4}$. Simplifying, $4 \times 20 = 80$, then 80 divided by 4 is 20. Alternatively, you could factor out the 4 in the numerator and cancel with the denominator, but in this case, the result is still 20. So that's a good way. Either way, multiplication and division are done from left to right.

Alternatively, let's check if there's any possibility that the entire numerator is being divided. Let's see. The entire expression is $\frac{4\times20}{4}$, so 4 × 20 is 80, then 80 divided by 4 is 20. So that's another way to see it, which is reassuring. So that seems consistent.

Another way to check: look at it by using properties of operations. But in this case, since the expression is written with multiplication and division, the order of operations tells us to do all calculations in the numerator first, then divide by the denominator. All steps lead to 20.

Another check: let me verify the multiplication. 4 × 20 = 80. Then 80 divided by 4 is 20. So no errors in the operations.

Hmm, I think that's solid. The answer should be 20. Let me just make sure there's no alternative interpretation. Let's write the original expression again:

Original expression: $\frac{(20-16)\times(12+8)}{4}$.
The numerator is 4 × 20 = 80, denominator is 4. So 80 divided by 4 is 20. Yep, exactly, that's correct.

Therefore, I'm confident that the answer is 20.
</think>

The result of $\frac{(20-16)\times(12+8)}{4}$ is calculated as follows:
1. **Simplify the parentheses inside the numerator:**
- $20 - 16 = 4$
- $12 + 8 = 20$
2. **Multiply the simplified results:**
- $4 \times 20 = 80$
3. **Divide by 4:**
- $\frac{80}{4} = 20$

**Final Answer:** 20

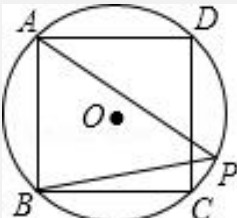

**Question:** In the given figure, a square ABCD is inscribed in a circle with center O. Point P is located on side CD. What is the value of angle APB?

___

**Show-o:**

The angle APB is 90 degrees.

___

**Emu3:**

The question seems to be a mix of letters and numbers, not a coherent question or statement. If you meant "is" or "are", the correct statement would be "The measure of angle APB is" or "The measure of angle APB is". If you meant "is" or "are", the correct statement would be "The measure of angle APB is" or "The measure of angle APB is".

___

**Janus Pro 7B:**

To find the measure of angle APB, we can use the properties of the square and the circle.
1. **Square ABCD**: The square is circumscribed around the circle, meaning all its sides are tangent to the circle. This implies that the diagonal of the square is equal to the diameter of the circle.
2. **Circle O**: The center of the circle is also the center of the square.
3. **Diagonals of a Square**: The diagonals of a square bisect each other and are equal in length.
4. **Angle APB**: Since ABCD is a square, angle APB is a right angle (90 degrees).

Therefore, the measure of angle APB is 90 degrees.

___

**MMaDA:**

<think>

To determine the measure of angle APB, we start by noting that square ABCD is circumscribed within circle O. This means that the circle is the circumcircle of the square, and the center of the circle is the same as the center of the square. Let's denote the center of the circle as O.

Since ABCD is a square, the diagonals AC and BD are equal in length and bisect each other at right angles. This means that angle AOB is 90 degrees because the diagonals of a square are equal and bisect each other at right angles.

Now, consider triangle APB. Since P lies on side CD, we can use the property of the circle that the angle subtended by a chord at the center is twice the angle subtended at any point on the circumference on the same side. In this case, AP and BP are chords of the circle meeting at point P on CD (which is opposite AB). Points A and B are at adjacent vertices, and P is on CD.

The key point is that when P moves along CD, angle APB is fixed, as AB is a fixed chord and P moves along the arc opposite AB. For a square inscribed in a circle, the arc AB subtends a right angle at any point on the circle opposite AB (i.e., on arc CD). Therefore, for any point P on CD, angle APB is always 45°. This follows because $\angle APB$ is half the degree of the arc AB, which is 90°, so:

$$\angle APB = \frac{1}{2} \times 90° = 45°$$

Thus, the measure of angle APB is $\boxed{45°}$.

</think>

The measure of angle APB is $\boxed{45°}$.

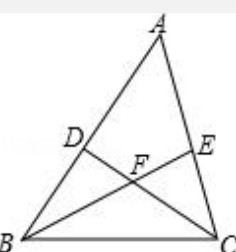

**Question:** In the diagram, triangle ABC is shown. Points D and E are located on segments AB and AC, respectively. Lines BE and CD intersect at point F. Given that angle BAC is 60°, angle DCA is 40°, and angle EBA is 30°, what is the measure of angle CFE?

---

### Show-o:
The angle measure of angle CFE is 60°.

---

### Emu3:
To solve for the degree measure of angle CFE, we can utilize the properties of triangles and angles. Since angle ABE measures 30°, angle ACD measures 40°, and angle ABC measures 60°, we can find angle ACD by subtracting angle A and angle ABE from 180°: 180° - 40° - 30° = 110°. Since triangle ABC is an isosceles triangle, angle BAC measures 40°. Therefore, angle CFE can be found by subtracting angle BAC from angle ABC: 40° - 110° = -30°. However, angles cannot have negative measures, so we take the positive value, giving us 60°. Therefore, the degree measure of angle CFE is 60°.

---

### Janus Pro 7B:
To find the measure of angle CFE, we can use the properties of triangles and the given angles. First, we know that angle ABE is 30°. Since angle ABE and angle ACD are adjacent angles in the same triangle, they are supplementary. Therefore, angle ACD is 180° - 30° = 150°.
Next, we know that angle A is 60°. Since angle A and angle ACD are adjacent angles in the same triangle, they are supplementary. Therefore, angle C is 180° - 60° = 120°.
Now, we can find angle CFE. Angle CFE is the supplement of angle C, so angle CFE = 180° - 120° = 60°.

Therefore, the degree measure of angle CFE is 60°.

---

### MMaDA:
<think>
To determine the measure of angle CFE, we need to analyze the given information and use the properties of triangles and angles.
1.  **Identify the given angles:**
- Angle A = 60°
- Angle ACD = 40°
- Angle ABE = 30°
2.  **Use the fact that the sum of the angles in a triangle is 180°:**
- In a triangle, the sum of the angles is 180°.  Therefore, we have:

$$60° + \angle ADC + 40° = 180°$$

$$60° + \angle AEB + 30° = 180°$$

3.  **Determine the measure of angle CFE:**
- Since angle AEB is an exterior angle to triangle CEF, it is equal to the sum of the two non-adjacent interior angles.  Therefore, angle AEB = angle ACD + angle CFE = 40° + angle CFE = 90°.
- Therefore, angle CFE = 90° - 40° = 50°.
Thus, the measure of angle CFE is 50 .
</think>

The measure of angle CFE is 50 .

---

## Qualitative Comparison of Multimodal Reasoning (3)

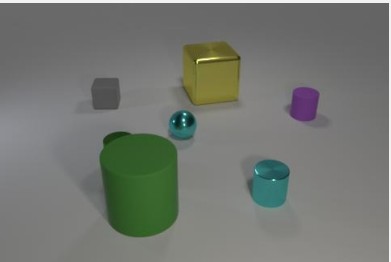

**Question:** What number of large things are the same material as the purple cylinder?

---

**Show-o:**

There are two large things that are the same material as the purple cylinder.

---

**Emu3:**

2

---

**Janus Pro 7B:**

There is one large object that shares the same material as the purple cylinder, which is the large green cylinder.
Answer: 1.

---

**MMaDA:**

```
<think>
To determine the number of items in the image, let's analyze the objects present:
1.  The large green cylinder is made of a material that appears to be rubber.
2.  The large yellow cube is made of a material that appears to be metal.
3.  The small gray cube is made of a material that appears to be rubber.
4.  The small cyan sphere is made of a material that appears to be metal.
5.  The small cyan cylinder is made of a material that appears to be metal.
6.  The small purple cylinder is made of a material that appears to be rubber.
The green cylinder is the only large object made of a material that is the same as the small
purple cylinder.  Therefore, there is 1 large object made of the same material as the purple
cylinder.
</think>

The number of large things that are the same material as the purple cylinder is 1.
```

## Qualitative Comparison of Multimodal Reasoning (4)

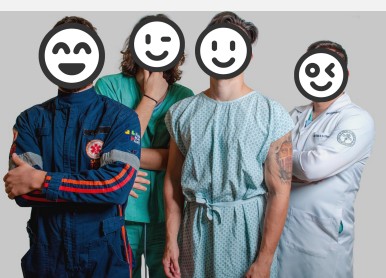

**Question:** Divide 24 apples equally among the people in the picture, how many apples can each person get?

---

**Show-o:**

In the image, there are four people, each with a different number of apples.  The person with the most apples has the most apples.

---

**Emu3:**

24.

---

**Janus Pro 7B:**
To divide 24 apples equally among the four people in the picture, you would divide 24 by 4.
24 / 4 = 6

Each person can get 6 apples.

---

**MMaDA:**
`<think>`
To determine how many apples each person can get, we need to divide the total number of apples by the number of people.
$24 \div 4 = 6$
So, each person can get 6 apples.
`</think>`

Each person can get 6 apples.

# G  Qualitative Examples (without Reasoning CoT)

MMADA can also efficiently handle general and simple tasks without generating CoT.

Qualitative Example of Text-to-Image Generation

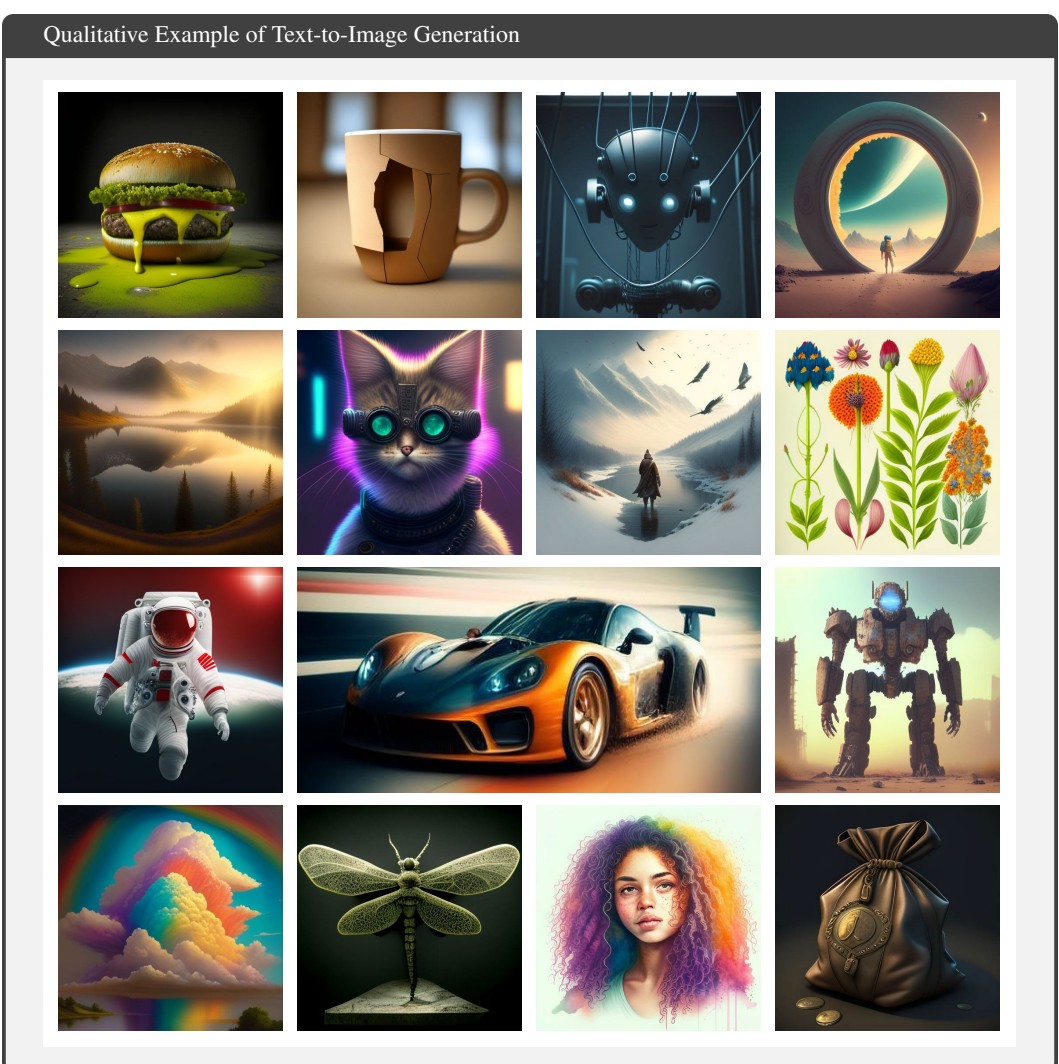

## Qualitative Example of Multimodal Understanding (1)

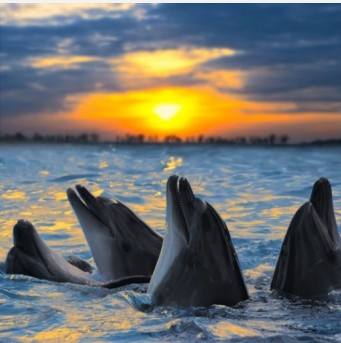

**Prompt:** Please describe this image in detail.

**MMaDA:**
The image captures a serene and picturesque scene of a group of dolphins swimming in the ocean during sunset. The sky is painted with hues of orange, yellow, and pink, transitioning into a deep blue as it meets the horizon. The sun is setting on the horizon, casting a warm glow that reflects off the water's surface, creating a shimmering effect. The dolphins are positioned in the foreground, with their bodies partially submerged in the water, their heads and upper bodies visible above the surface. The water appears calm, with gentle ripples around the dolphins, adding to the tranquil atmosphere. The overall composition of the image conveys a sense of peace and natural beauty.

## Qualitative Example of Multimodal Understanding (2)

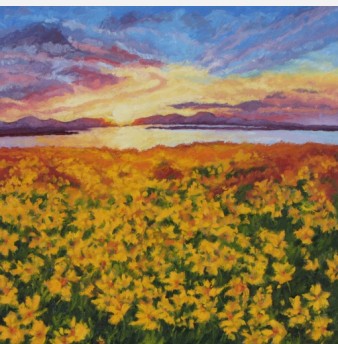

**Prompt:** Please describe this image in detail.

**MMaDA:**
The image is a vibrant and colorful landscape painting that captures the essence of a sunset over a field of yellow flowers. The sky is a stunning blend of warm, soft hues, transitioning from deep purples and blues at the top to warm oranges and yellows near the horizon. The sun is positioned low on the horizon, casting a golden glow that illuminates the sky and the field below. The field of yellow flowers, likely sunflowers, stretches across the foreground, creating a sea of color that contrasts beautifully with the cooler tones of the sky. The painting is executed with broad, expressive brushstrokes, giving it a dynamic and lively feel. The overall composition is balanced and harmonious, capturing the tranquility and beauty of the natural world.

## Qualitative Example of Multimodal Understanding (3)

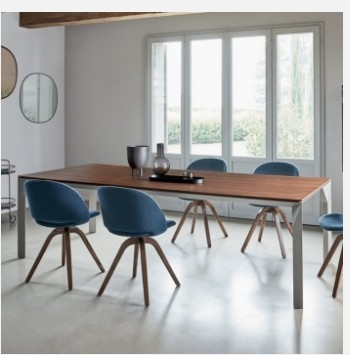

**Prompt:** Please describe this image in detail.

**MMaDA:**
The image depicts a modern dining area with a minimalist aesthetic. The centerpiece is a long dining table with a sleek, dark finish, supported by four sturdy legs. The table is set with a few items, including a black vase, a few books, and a small decorative object, adding a touch of elegance to the space. The table is surrounded by four blue velvet chairs with wooden legs, providing a striking contrast to the dark tones of the table. The room has a clean, open feel with light-colored walls and large windows that allow natural light to flood in, creating a bright and airy atmosphere. The left is a round mirror with a simple frame, enhancing the modern look of the room.

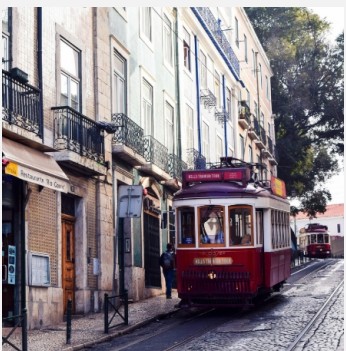

**Prompt:** Please describe this image in detail.

MMaDA:
The image depicts a charming, narrow street in a European city, likely in, given the architectural style and the presence of a red tram car. The street is lined with tall, narrow buildings painted in soft pastel colors, including shades of pink, green, and beige. The buildings have ornate balconies and shutters, adding to the quaint charm of the scene. The street is paved with cobblestones, and there tram tracks run down the center, indicating that this is a tram lane. A red tram car is parked on the right. The tram is stationary, and the overall atmosphere is serene and picturesque.

