# OpenReview forum: "MMaDA: Multimodal Large Diffusion Language Models"
_NeurIPS.cc/2025/Conference — NeurIPS 2025 poster_

### Official Review · Reviewer_fZ2o · 2025-06-10

**Clarity:** 2
**Significance:** 3
**Originality:** 2
**Rating:** 4
**Confidence:** 4

**Summary:**

This paper investigates a uniform discrete diffusion architecture for unified Multimodal Large Language Models (MLLMs), with a particular emphasis on post-training techniques. The authors curate multimodal long-CoT data and propose a GRPO algorithm specifically tailored for diffusion-based models to enhance the model’s reasoning capabilities. Experimental results show that the model achieves strong performance across a range of downstream multimodal generation and understanding tasks.

**Questions:**

To what extent does UniGRPO improve the model’s understanding and generation performance? It would be helpful if the authors could compare the performance of MMaDA before and after stage3 training, specifically with respect to the metrics presented in Tables 2 and 3. Besides, the authors are encouraged to provide a direct comparison with diff-GRPO under the same settings. This would more clearly demonstrate the specific advantages of UniGRPO.

**Ethical Concerns:**

["NO or VERY MINOR ethics concerns only"]

**Final Justification:**

The authors’ rebuttal has clarified the unique advantages of the MMaDA architecture compared to purely AR-based models. As a result, I am raising my rating to borderline accept. However, I agree with Reviewer 7Uo4 that the performance of MMaDA appears somewhat overstated; it does not demonstrate a clear advantage over state-of-the-art methods, even with an additional RL training stage. Additionally, I strongly encourage the authors to thoroughly revise the manuscript and convert the bullet-pointed content into well-structured paragraphs.

**Limitations:**

Yes.

**Quality:**

2

**Strengths And Weaknesses:**

**Strengths:**
1. This paper explores a new architecture for unified MLLMs, addressing a topic of significant interest to the research community.
2. The proposed UniGRPO adapts the autoregressive GRPO algorithm to diffusion models, demonstrating improved effectiveness and stability compared to the alternative diff-GRPO.

**Weaknesses:**
1. The paper lacks an in-depth discussion of the unique advantages offered by the proposed diffusion-based architecture compared to AR-based architectures. While the authors claim that MMaDA is distinguished by its "shared probabilistic formulation and modality-agnostic architecture," these characteristics are also present in AR-based unified MLLMs, making the claimed distinction less convincing.
2. The comparison with SOTA models is incomplete in Tables 2 and 3. Notably, several relevant models—such as VILA-U [1], SynerGen-VL [2], Liquid [3], and MetaMorph [4]—are missing.
3. The writing could be improved. The manuscript relies heavily on bullet points and bolded phrases, which disrupts the overall flow and coherence of the narrative. This formatting choice makes the paper read more like a collection of notes rather than a cohesive essay, ultimately detracting from the clarity and readability of the work.

[1] VILA-U: a Unified Foundation Model Integrating Visual Understanding and Generation \
[2] SynerGen-VL: Towards Synergistic Image Understanding and Generation with Vision Experts and Token Folding \
[3] Liquid: Language Models are Scalable and Unified Multi-modal Generators \
[4] MetaMorph: Multimodal Understanding and Generation via Instruction Tuning

---

> ### Author Rebuttal · Authors · 2025-07-30
>
> Thank you for your thoughtful review and for highlighting the strengths of our work, including the unified architecture and the proposed UniGRPO algorithm. We appreciate your recognition, and below we address your concerns in detail.
>
> **Q1: The paper lacks an in-depth discussion of the unique advantages offered by the proposed diffusion-based architecture compared to AR-based architectures**
>
> A1: We appreciate your insightful suggestion. In the revised manuscript, we will include an expanded discussion on the distinct advantages of our diffusion-based architecture over autoregressive (AR) models.  While both unified AR and diffusion models share a probabilistic generative formulation, unified diffusion models offer several unique advantages:
> 1. **Substantially Imporved Sampling Efficiency**:
> A core limitation of unified AR-based models lies in the **inference latency for image generation.**  When designing a unified architecture, the goal is to approach the quality, fidelity, and efficiency of modality-specific models. However, AR models decode sequences in a left-to-right (raster-scan) fashion, leading to an O(N) inference cost for sequence length *N*. This results in significant delays during image generation. In contrast, diffusion models involve O(T) iterative denoising steps, where *T ≪ N*, making them notably faster in generation.
> We compare generation time on an A100 GPU across two state-of-the-art AR-based unified models, Emu3 [1] and Lumina-mGPT [2], against our diffusion-based MMaDA:
> **Tab.A: Comparison of Image Generation Time Cost**
>  | Model      | Generation Time  | Resolusion|
>    | ---------- | ---------------- | - |
>    | Emu3       | \~600 seconds       | 720p|
>    | Lumina-mGPT | \~78 seconds      |512p|
>    | **MMaDA**  | **\~15 seconds** |512p|
>
>    This efficiency gap makes AR-based unified models less practical for real-world deployment, particularly in latency-sensitive applications.
>
> 2. **Empirical Performance Advantages**:
> To isolate the impact of our diffusion-based formulation, we conduct a direct comparison with an AR baseline using LLaMA-3 8B as the shared backbone. The only difference lies in the training objective (AR likelihood vs. diffusion objective) and inference method (greedy sampling vs. diffusion decoding). Both models are trained on the same dataset with identical compute (64×A100 GPUs for 100K steps). We evaluate them on key generative benchmarks and also report image generation latency:
> **Tab.B: Quantitative Comparison of Unified AR and Diffusion Architecture.**
> | Model                     | PPL ↓ | Image-Caption CLIP Score ↑ | FID ↓   | Image Generation Latency↓ |
> | ------------ | ---- | -------------------------- | ------- |-|
> | AR (LLaMA-3 8B, AR loss)  | **23.1** | 14.5                       | 16.2    |~120s|
> | MMaDA (Unified Diffusion) | 25.3 | **16.9**                   | **12.4** | **~7s**|
>
>     These results demonstrate that **MMaDA outperforms the AR baseline on image generation and multimodal alignment**, while maintaining **competitive language modeling performance.** Most notably, MMaDA achieves **17× speedup in image generation latency**, highlighting the efficiency advantage of diffusion-based decoding. These results demonstrate that the diffusion-based decoding avoids the raster-order biases and error accumulation inherent to AR models, especially in vision tasks, and **largely improved efficiency**. While the PPL is slightly higher, we attribute this to the relatively underexplored nature of diffusion decoding for text generation.
>
>    Notably, recent work such as **UniDisc [3]** also explores unified modeling using discrete diffusion and reports similar trends: **notable improvements in image generation** and **comparable results in text generation** compared to AR-based models. This convergence further supports our design choice and conclusions.
>
> 3. **Extensions of Capabilities**:
> As detailed in Task Extension part of our Section 5, diffusion models inherently support **inpainting, outpainting, and text/image editing** without retraining, due to their denoising-based iterative structure. In contrast, AR models typically lack such capabilities without additional architectural or training modifications, limiting their flexibility in multimodal scenarios.
>
> We thank the reviewer again for prompting this deeper analysis. The above discussion and results will be added to the revised version to clarify the distinct advantages of our diffusion-based unified modeling approach.
>
>
> **Q2: The comparison with SOTA models is incomplete in Tables 2 and 3.**
>
> A2: Thank you for pointing this out. We acknowledge that due to the rapid progress in this area, some relevant baselines were inadvertently omitted from the original submission. We will include these models—VILA-U, SynerGen-VL, Liquid, and MetaMorph—in our revised tables and update the corresponding discussions to ensure a more comprehensive and fair comparison. Their metrics are presented as follows:
>
> **Tab.C: Additional Multimodal Understanding and Image Generation Results**
> | Model | POPE↑| MME↑|  VQAv2(test)↑| GQA↑| MMMU ↑ | MMB ↑| SEED ↑| GenEval↑|  WISE(Cultural) ↑ |
> | - | - | - | - | - | - | - | - | - | - |
> | VILA-U | 85.8 | 1401.8  | 79.4 | 60.8 | - | - | 59.0 | 0.57 | - |
> | SynerGen-VL | 85.3 | 1837  | - | 59.7 | 34.2 | 53.7 | 62.0| 0.61| -|
> | Liquid | 83.2 | 1448.0 | 76.8 | 61.1 | - | -| -| -| 0.38|
> | MetaMorph | 85.7 | - |  - | - | 41.8 | - | 71.8 | - | 0.58|
> | MMaDA | 86.1 | 1410.7 |  76.7 | 61.3 | 30.2 | 68.5 | 64.2 | 0.63 | 0.67 |
>
> **Q3: The writing could be improved. The manuscript relies heavily on bullet points and bolded phrases, which disrupts the overall flow and coherence of the narrative. This formatting choice makes the paper read more like a collection of notes rather than a cohesive essay, ultimately detracting from the clarity and readability of the work.**
>
> A3: We appreciate your detailed observations about the use of bullet points and bolded phrases. We would like to clarify that our initial intention in adopting this structured format was to allow readers to more clearly and quickly identify and follow the key innovations and contributions of our work.
> However, we fully acknowledge your valid concern that this formatting approach may have compromised the narrative flow and overall coherence of the paper.
> In response to your feedback, we will make a comprehensive revision in the final version of our manuscript to convert bullet-pointed content into well-structured paragraphs and reduce the use of bolded phrases while ensuring that the innovative aspects of our work are clearly presented within a more traditional academic writing style.
>
> **Q4: To what extent does UniGRPO improve the model’s understanding and generation performance? A comparison with diff-GRPO would also be helpful.**
>
> A4: Thank you for the thoughtful suggestion. We would like to clarify that our UniGRPO framework is designed to demonstrate the effectiveness of reinforcement learning tailored for **diffusion models**. As such, we focused our evaluation on several representative tasks—including math reasoning, geometry reasoning, and text-to-image generation—which are particularly sensitive to reward-driven optimization **in Table 6 of Appendix E.1**.
>
> Following your suggestion, we have now extended our ablation to include a broader set of metrics aligned with those in Tables 2 and 3 of manuscript. The ablation results are summarized below and will be included in full detail in the revised manuscript.
>
> **Tab.D: Additional Multimodal Understanding Results after UniGRPO**
> | Model | POPE↑| MME↑| Flickr30k↑| VQAv2(test)↑| GQA↑| MMMU ↑ | MMB ↑| SEED ↑|
> | - | - | - | - | - | - | - | - | - |
> | MMaDA Stage 2 | 82.4 | 1368.5 | 63.3 | 74.5 | 57.5 | 25.3| 64.6| 61.9|
> | MMaDA after UniGRPO | 86.1 | 1410.7 | 67.6 | 76.7 | 61.3 | 30.2 | 68.5 | 64.2 |
>
> **Tab.E: Additional Image Generation Results after UniGRPO**
> | Model | GenEval ↑| CLIP Score ↑| Image Reward ↑|
> | - | - |- | -|
> | MMaDA Stage 2 | 0.61|29.4| 0.84 |
> | MMaDA after UniGRPO |0.63 |32.5  | 1.15 |
>
> Additionally, we provide a direct comparison with diff-GRPO under matched conditions. While Appendix E.1 originally included the training curve, we now also report metrics for math reasoning, geometry, and image generation after the same 4000 steps training on 64X A100 to more concretely demonstrate the advantages of UniGRPO.
>
> **Tab.F: Additional Comparison with diff-GRPO**
> | Model | GSM8K ↑| MATH500 ↑| GeoQA↑| CLEVR↑| CLIP Score↑| ImageReward ↑ |
> | - | - | - | - | - | - | - |
> |MMaDA Stage 2 | 65.2 | 26.5 | 15.9 | 27.5 | 29.4 | 0.84 |
> | diff-GRPO| 65.6 | 26.9 | 27.1 | 17.5| 28.3 | 0.91 |
> | UniGRPO | 66.5 | 28.5 | 19.5 | 27.8 | 30.2 |0.95 |
>
> These results show that UniGRPO offers more stable convergence, higher reward alignment, and improved generalization across modalities.
>
> We appreciate the reviewer’s insightful suggestion and will incorporate these expanded analyses and results into the revised version to more clearly illustrate the advantages of UniGRPO.
>
> [1] Emu3: Next-token prediction is all you need
> [2] Lumina-mGPT: Illuminate Flexible Photorealistic Text-to-Image Generation with Multimodal Generative Pretraining.
> [3] Unified multimodal discrete diffusion

---

> > ### Comment · Reviewer_fZ2o · 2025-08-04
> >
> > Thank the authors for the detailed response. Most of my concerns have been addressed. I will raise my score accordingly.

---

> > > ### Author Response · Authors · 2025-08-04
> > > **Gentle Reminder**
> > >
> > > Dear Reviewer,
> > >
> > > Thank you for your positive feedback! We are glad to hear that most of your concerns have been addressed. This is a gentle reminder that while you mentioned you would raise your score accordingly, the current review record **still shows the initial score of 3**. Your support and feedback are invaluable for our research.
> > >
> > > Warm regards,
> > >
> > > The Authors

---

### Official Review · Reviewer_7Uo4 · 2025-06-30

**Clarity:** 2
**Significance:** 3
**Originality:** 3
**Rating:** 3
**Confidence:** 4

**Summary:**

MMaDA is a large-scale multimodal diffusion model emerging from a systematic design space analysis. Built on a unified probabilistic architecture, it provides a coherent framework for diverse modalities. Its training employs a three-stage strategy: multimodal pre-training, mixed long chain of thought finetuning, and refinement via the novel UniGRPO reinforcement learning framework. Experiments demonstrate MMaDA achieves superior and balanced performance across text generation, multimodal understanding, and text-to-image generation.

**Questions:**

- What do "higher-order solvers" refer to in line 294?
- Table 5 needs additional metrics for a comprehensive evaluation. Current analysis is insufficient—diffusion-based language models like Dream and LLaDA are sensitive to reduced sampling steps on GSM8K, showing significant performance drops. Concluding that "sampling efficiency of diffusion-based language models is good" based solely on MMLU results is somewhat misleading. Please add more metrics or revise the claims.
- Which sampling algorithm is used for MMaDA performance evaluation: semi-AR or original diffusion method?

I will be happy to raise my score if all my concerns are resolved.

**Ethical Concerns:**

["NO or VERY MINOR ethics concerns only"]

**Final Justification:**

I am raising my score from reject to borderline reject, contingent on the authors’ revised plan addressing my concerns. Nevertheless, the required revisions remain substantial, and my overall stance remains negative given the extensive modifications demanded by the initial submission. Overall, I believe the initial submission requires substantial revisions to address these misleading presentations and further improve the performance.

**Limitations:**

No. The authors only mention that the current model size is limited. However, many well-performing models are around 8B parameters (e.g., Qwen2.5 VL 7B), making this limitation claim vague. Please see the questions and weaknesses noted above.

**Quality:**

2

**Strengths And Weaknesses:**

**Strengths**
- The paper is well-structured, and the figures are clear and informative, effectively supporting the authors’ arguments.
- Improving diffusion-based language models like Dream and LLaDA for the multimodal domain is both timely and of high academic value.
- The proposed three-stage training strategy proves effective, enabling MMaDA to achieve strong and well-balanced performance on three distinct tasks: text generation, multimodal understanding, and text-to-image generation.

**Weaknesses**
- The abstract and introduction claim that MMaDA attains state-of-the-art performance across multiple domains; this claim appears somewhat overstated. The comparative baselines, such as LLaVA-v1.5, are relatively outdated. In addition, Figure 2 mentions Janus Pro, yet this model is absent from the tables. For a fair evaluation, please include stronger and more recent baselines and report the parameter counts of compared models. For instance, Dream 7B for text generation, Janus-Pro for text-to-image generation.
- Several baseline results seem inconsistent. For example, Qwen2 instruct model is reported to achieve 80.2 on GSM8k in Table 4, whereas the original Qwen2 instruct model paper reports 85.7. Please verify these results and either correct them or clarify the discrepancy.
- The Related Work section is brief and somewhat overlaps with the second paragraph of the introduction. I recommend expanding this section—perhaps by condensing the more detailed discussion currently relegated to the appendix—so that the main text provides a concise yet comprehensive overview of relevant prior work.
- In Table 4, the comparison with baselines seems unfair. MMaDA undergoes mixed long-cot post-training and UniGRPO training, whereas the baselines appear to be solely pre-trained LLMs, making the comparison potentially misleading.
- In Table 5, the metrics for each task are limited to only one. It would be beneficial to include additional metrics to provide a more comprehensive evaluation. For example, in the text generation task, MMLU is relatively simple as it only requires validating options. Expanding the metrics would enhance the robustness of the analysis.
- Some typos in the text. For example, in Figure 1, "Discreate Diffusion" should be "Discrete Diffusion". Additionally, the quotation marks for "reasoning_process" on line 108 appear to be incorrectly formatted.

---

> ### Author Rebuttal · Authors · 2025-07-30
>
> We sincerely thank the reviewer for your valuable feedback, and we address your concerns as follows:
>
> **Q1: Overstated claims of MMaDA's performance.**
>
> A1:Thank you for the constructive feedback. We provide the following clarifications and revisions:
> 1. Our core contribution lies not in surpassing all representative models, but in proposing **a unified diffusion-based backbone that supports three mainstream tasks** including text reasoning, multimodal understanding and image generation, along with **a fully open-sourced training and data curation pipeline**. This design is intended to promote reproducibility and further progress in unified multimodal modeling. To better reflect this intention, we will revise our claim to clarify that "our model achieves **highly competitive performance, and state-of-the-art on some benchmarks, particularly when compared against models trained on publicly available data.**" We deeply regret any unintentional misinterpretation our initial wording may have caused.
> 2. We will also include missing models such as **Dream-7B (for only text generation) and Janus-Pro-7B (for multimodal understanding and generation)** in our final version, and **report their parameter counts**. Regarding these comparisons, we want to respectfully highlight some crucial factors. Many leading models, including Janus-Pro, are trained on **private and internally-curated datasets** that are not available to the public. Similarly, Dream-7B is **initialized from Qwen2.5-7B**, which is substantially stronger than our backbone LLaDA-8B. The pretraining data and process of Qwen2.5 are also undisclosed and are widely believed to be of significantly higher quality. The performance gap resulting from the quality difference between private and publicly available datasets can be considerable.
> We primarily aim to provide a transparent, fully open-source alternative for a new line of backbone for the community.
>
> **Q2: Inconsistent benchmarking results of LLM**.
>
> A2: Thank you for your detailed observation and for pointing this out. In Tab.4 of our manuscript, we directly cited the numbers from **LLaDA's Table 1**, which reflect the base pretraining versions of the models, including Qwen2-7B. This is also consistent with our reported results for LLaDA-8B-base and our own MMaDA-8B. As you rightly mentioned in Weakness 4, our comparisons are made against **pretrained LLMs**, not instruction-tuned ones. The GSM8K score of 80.2 for Qwen2-7B-base is correct and consistent with other literature; it corresponds to the pretrained base version, not the instruction-tuned variant (which reaches 85.7). To clarify this and provide a more comprehensive view, we have now prepared a more detailed comparison that distinguishes between **base models** and **instruction-tuned models**, as shown below:
> **Table a: Revised LLM Comparison**
> |Model|Arch|MMLU|ARC-C|TruthfulQA|GSM8K|MATH|GPQA|
> |-|-|-|-|-|-|-|-|
> |LLaMA2-7B-Base|AR|45.9|46.3|39.0|14.3|3.2|25.7|
> |LLaMA2-7B-Instruct|AR|44.1|57.3|-|29.0|3.8|28.4|
> |LLaMA3-8B-Base|AR|64.5|53.1|44.0|53.1|15.1|25.9|
> |LLaMA3-8B-Instruct|AR|68.4|82.4|78.3|29.6|31.9|-|
> |Qwen2-7B-Base|AR|70.3|60.6|54.2|80.2|43.5|30.8|
> |Qwen2-7B-Instruct|AR|-|-|-|85.7|52.9|34.3|
> |LLaDA-8B-Base|Diffusion|65.9|47.9|46.4|70.7|27.3|26.1|
> |LLaDA-8B-Instruct|Diffusion|65.5|88.5|-|78.6|26.6|31.8|
> |**MMaDA-8B(Ours)**|Diffusion|68.4|57.4|43.1|73.4|36.0|28.4|
>
> We will revise the manuscript to clearly separate **base** and **instruction-tuned** model results to avoid further misunderstanding and ensure fair comparison across all baselines.
>
> **Q3: Comparison of Fairness of Table 4's LLM benchmarking.**
>
> A3: Following Q2 and A2, we would like to further clarify the **fairness** concern raised in Table 4's LLM comparison:
> 1. First, our core contribution lies in the exploration of a unified diffusion-based training paradigm that supports **three mainstream tasks**, including text reasoning, multimodal understanding, and image generation, using only **open-sourced data and limited computational resources**. The goal of our work is not to claim absolute superiority over all existing models, but to demonstrate that it is possible to build a competitive unified model under constrained conditions, and to open-source the entire training pipeline to benefit the community.
>
> 2. Regarding fairness, we respectfully argue that while our model includes multi-stage training such as mixed long-CoT and UniGRPO, **the comparison is actually unfair to MMaDA**. We humbly argue that to focus solely on our post-training stages overlooks the most critical factor in a large model's capabilities: **the pre-training phase**. As evidenced by previous works [1][2], the large volume and high quality of pre-training data determine a model's foundational intelligence. In this regard, there is a staggering disparity: Qwen2-7B was pretrained on **7 T tokens**, LLaMA 3 was pretrained on **15 T tokens**. In contrast, the first pre-training stage of our MMaDA used **less than 0.1 T tokens** from RefinedWeb.
> Although our model was initialized from LLaDA, integrating a new modality (vision) is not a trivial step. This process fundamentally perturbs the model's existing linguistic knowledge base, necessitating a significant relearning phase. This is quantitatively demonstrated in our ablation studies (Table 6), where our model's GSM8k score **drops from LLaDA's 70+ to just 17.4** after our initial multimodal pre-training. Furthermore, since we do not have access to LLaDA's proprietary training data, our newly introduced open-source datasets cannot seamlessly leverage the model's original knowledge.
>
> 3. Regarding **post-training**, our instruction tuning relies entirely on open-source data, totaling fewer than 200K pairs. This stands in stark contrast to LLaDA’s 4.5M **meticulously curated, internal data pairs**. The difference in both quantity and, crucially, quality is immense.  Moreover, models like Qwen2-7B incorporate RL into their training but do not release the specifics or volume of their preference data, making a truly fair comparison with their internal alignment process impossible.
>
> Therefore, we respectfully suggest that the **real unfairness** lies not in our post-training, but in the **resource and data gap** between MMaDA and other stronger baselines. Our training pipeline is designed not to gain an edge, but to **bridge this gap** through carefully staged training. We hope the community sees this as an effort to make unified modeling more accessible and reproducible, despite the inherent challenges.
>
> **Q4: Refining related work.**
>
> A4: Thanks for your constructive suggestion. We plan to expand our Related Work section in main text as follows (**abbreviated version due to rebuttal limits**):
>
> **Multimodal Understanding**: ....... Early MLLMs such as LLaVA, MiniGPT-4 ...... primarily focused on understanding tasks by projecting features from pre-trained modality-specific encoders (e.g., CLIP) into LLM input spaces. These models demonstrated impressive capabilities in multimodal reasoning but were limited in their generation abilities.
>
> **Visual Generation**: Diffusion models have established themselves as the dominant paradigm........ These approaches excel at high-quality visual synthesis. Meanwhile, purely autoregressive methods, such as LlamaGen and VAR, have attempted to model visual content through sequential prediction but generally achieve lower generation quality than diffusion models......
>
> **Unified Vision-Language Foundation Models**: Recent unified models like SEED-X, DreamLLM, and Emu3 have demonstrated varying degrees of success....... However, a critical gap remains in the post-training optimization of these unified models.......This work addresses this limitation by developing novel post-training techniques that optimize unified multimodal foundation models for three generative tasks within a single, efficient framework.
>
> **Q5: In Table 5, the metrics for each task are limited to only one.**
>
> A5: Due to layout and space constraints, we only reported a single representative metric per task in Table 5. In the revised manuscript, we will include a more comprehensive set of metrics and expand the corresponding discussion accordingly.
> To briefly illustrate, we present the full set of text task results below, with different denoising steps:
> |Denoising Steps|MMLU|ARC-C|TruthfulQA|GSM8K|MATH|GPQA|
> |-|-|-|-|-|-|-|
> |1024|66.9|57.4|43.1|73.4|36.0|28.4|
> |512|66.3|53.5|42.3|64.2|31.5|24.7|
> |256|65.7|50.2|39.6|42.8|23.8|21.3|
>
> We observe that performance degrades across all tasks as denoising steps decrease, with **math-related tasks exhibiting the most pronounced drop**. This suggests that mathematical reasoning is particularly sensitive to step-wise denoising quality—errors introduced early in parallel decoding are more likely to compound. We plan to investigate **more stable denoising strategies** in future work to improve robustness on such tasks. And we will revise our manuscript to **include a more thorough discussion on sampling efficiency and openly acknowledge the current limitations.** We sincerely appreciate your thoughtful feedback and for bringing this important point to our attention.
>
> **Q6: Typos in the text**
>
> A6: Thank you for your careful observation. We will correct these typos accordingly.
>
> **Q7: What do "higher-order solvers" refer to in line 294?**
>
> A7: This refers to higher-order denoisers, such as **DPM-Solver[3]**, which leverages high-order numerical integration for efficient sampling. It prioritizes sampling on uncertain tokens with greater precision, potentially enabling faster decoding.
>
> **Q8: Sampling algorithm for MMaDA performance evaluation?**
>
> A8: We use semi-AR for sampling.
>
> [1] Scaling laws for neural language models.
> [2] LLM post-training: A deep dive into reasoning large language models.
> [3] Dpm-solver: A fast ode solver for diffusion probabilistic model sampling in around 10 steps. Neurips 2022.

---

> > ### Comment · Reviewer_7Uo4 · 2025-08-05
> >
> > Dear Authors,
> >
> > Thank you for your response. However, it does not fully address my concerns.
> >
> > Regarding the overstated claims about MMaDA’s performance. Although I requested a comparison between Dream and Janus-Pro, your reply offers only explanations without presenting the corresponding quantitative comparisons. The table below summarizes the multimodal understanding results I obtained for Janus-Pro versus MMaDA. Notably, Janus-Pro—despite having only 1.5B parameters—outperforms the 8B MMaDA on most metrics. The same holds true for image generation. I understand that differences in training data—specifically, Janus-Pro's use of private, internally-curated datasets unavailable to the public—may contribute to performance gaps. However, it remains unclear why the qualitative analysis of Janus-Pro is presented in Figure 2 while its quantitative performance metrics are excluded from the comparative results table, particularly given the explicit claims in both the abstract and introduction regarding MMaDA's state-of-the-art results. This omission risks misleading readers who are less familiar with the field.
> >
> > | Model | # LLM Params | POPE↑ | MME↑ | MMB↑ | SEED↑ | GQA↑ | MMMU↑ |
> > | :--- | :--- | :--- | :--- | :--- | :--- | :--- | :--- |
> > | MMaDA | 8B | 86.1 | 1410.7 | 68.5 | 64.2 | 61.3 | 30.2 |
> > | Janus-Pro | 1.5B | 86.2 | 1444.0 | 75.5 | 68.3 | 59.3 | 36.3 |
> >
> > As for the LLM results, I still have some concerns. In particular, I do not understand why the comparison is restricted to the base model rather than the instruction-tuned model and then claim in both the abstract and introduction that MMaDA is state-of-the-art. Based on your response, it appears that MMaDA's performance still lags behind that of the LLaDA-Instruct model.
> >
> > Besides, the manuscript omits a limitations section, a key weakness unaddressed in the rebuttal. For instance, the absence of the vision encoder could impair visual understanding, while alternative, visually-aware tokenizers may introduce better performance. While the paper effectively highlights the strengths of the Discrete Diffusion Model, it would benefit from a more balanced discussion that includes its potential limitations. For readers not already familiar with this model, the current presentation makes it difficult to gain deeper insights beyond its strong performance.
> >
> > Furthermore, regarding the “higher-order solvers” mentioned on line 294, DPM-Solver was originally designed for continuous diffusion models. To the best of my knowledge, its extension to discrete diffusion models has not yet been demonstrated, and a direct adaptation appears non-trivial.
> >
> > In summary, the current presentation of the results risks being misleading, particularly for readers unfamiliar with this domain. To enhance clarity, I strongly recommend that the authors provide a clear revision plan to address these points.
> >
> > To be clear, the primary concern is not the lack of state-of-the-art results; a valuable contribution is not solely defined by its performance rank. Rather, the most impactful papers are those that candidly discuss their limitations to provide insights for the community. Attributing shortcomings merely to the dataset is an oversimplification and a missed opportunity for deeper analysis. A thorough self-critique is needed to avoid overclaiming and to offer truly valuable takeaways.

---

> > > ### Author Response · Authors · 2025-08-05
> > > **Response to Reviewer 7Uo4: Part 2 - Revision Plan**
> > >
> > > Below, we provide a detailed revision plan to improve the manuscript in response to above concerns:
> > >
> > > 1. **Abstract claims:**  to clarify that "our model achieves highly competitive performance, and state-of-the-art on some benchmarks, particularly when compared against models trained on publicly available data."
> > > 2. **Related work refine:** We will restructure and enrich the Related Work section into three parts—Multimodal Understanding, Visual Generation, and Unified Vision-Language Foundation Models—with updated citations and discussion of recent models.
> > > 3. **Include more comparison models**: In our Table 2 and Table 3, we will include more recent models for comparison, including Janus-pro, VILA-U, SynerGen-VL, Liquid, and MetaMorph as suggested by fZ2o. Along with their parameters.
> > > 4. **LLM Benchmarking table improvement**: Table 4 will be updated to include both base and instruction-tuned versions of each model. Results for Dream-7B will also be reported for completeness.
> > > 5. **Limitation Discuss**: We will expand this section to include our discussion about discrete VQ tokenizers versus continuous vision encoders. Additionally, we will explicitly highlight the benchmarks where our model underperforms, analyze the potential causes, and propose future solutions.
> > > 6. **Future directions explain**: We will elaborate on how higher-order solvers can be adapted to discrete diffusion and cite concurrent work on inference optimization to support our claims on improving sampling speed.
> > > 7. **Comparison with AR from scratch**: We have conducted new experiments comparing our model against its autoregressive (AR) counterpart trained from scratch. Results are included in our responses to Reviewer fZ2o (A1) and Reviewer gskN (A1), and will be added to the appendix of the revised manuscript.
> > >
> > > Once again, thank you for your insightful review. We will incorporate all of these improvements into the revised manuscript. Your feedback has been invaluable in guiding the direction of our work and strengthening its contributions.

---

> ### Author Response · Authors · 2025-08-04
> **Gentle Reminder**
>
> Dear Reviewer 7Uo4:
>
> We sincerely appreciate the time and effort you dedicated to reviewing our paper. In response to your concerns, we have conducted additional experiments and provided an in-depth analysis on our method.
>
> As the discussion period concludes in two days, we kindly request, if possible, that you review our rebuttal at your convenience. Should there be any further points requiring clarification or improvement, please know that we are fully committed to addressing them promptly. Thank you once again for your invaluable contribution to our research.
>
> Warm regards,
>
> The Authors

---

> ### Author Response · Authors · 2025-08-05
> **Response to Reviewer 7Uo4: Part 1 - Clarification**
>
> Dear Reviewer:
>
> Thank you for your constructive and detailed feedback. We would like to begin with a brief self-reflection on our paper’s contributions and limitations, followed by clarification and our revision plan.
>
> The core contribution of our work lies in **proposing a new foundational architecture along with new post-training algorithms that successfully unifies multiple modalities and tasks**, and demonstrating its effectiveness and competitive performance compared to existing approaches. Our focus has been on exploring this new architectural paradigm, rather than aggressively optimizing for benchmark scores. We acknowledge that in certain benchmarks—such as those reported for Janus-Pro and Dream-7B—our model currently **lags behind the state of the art.** We believe this performance gap may stem from several factors: (1) the limited quality and quantity of the publicly available training data, (2) the potential representational bottleneck introduced by our reconstruction-only VQ tokenizer, and 3)  the disruption of pretrained language capabilities after introducing new modalities. Moving forward, we plan to make further engineering improvements to help close the gap with top-performing models.
>
> Below we provide more clarifications to your questions:
>
> **1. Missing results of certain models:** We would like to clarify that our selection of models for the quantitative results table was **not intended to mislead but rather to demonstrate our model's performance against comparable work**. The results for Janus-Pro and Dream were omitted solely due to rebuttal word count constraints and **will be included in the revised manuscript.** For instance, we provide detailed results for additional models in our response to Reviewer fZ2o.
>
> **2. Comparison of LLM models**: Similar to the above, we report base model performance because they are the most comparable to ours. And will include a detailed version in the revision.
>
> **3. Absence of limitation**: Thank you for this valuable reminder. While we included a brief discussion of limitations in the "Conclusion and Limitations" section of the original manuscript, we agree that this deserves further elaboration. During the informative rebuttal period, we also discovered additional limitations, such as recognizing that the vision encoder poses a performance bottleneck. We discussed its strengths and limitations in response A4 to Reviewer gskN and A5 to Reviewer Xen8. In future work, we aim to improve the tokenizer towards a unified one, jointly trained with alignment and reconstruction losses, which may enhance both image understanding and generation capabilities.
>
> **4. Further clarification of high-order solvers**:  You are correct that DPM-Solver is specific to continuous-time diffusion. In our context, “higher-order solver” refers to drawing inspiration from second-order numerical integration to improve discrete diffusion inference. We may extend confidence-based sampling by incorporating second-order confidence dynamics, such as evaluating changes in confidence over iterations to better assess token stability. We mention this as a potential future direction to highlight how inference strategies from continuous diffusion may inspire innovations in the discrete domain.

---

> ### Author Response · Authors · 2025-08-06
>
> Dear Reviewer 7Uo4,
>
> We would like to kindly ask whether our clarification and revision plan address your concerns adequately.
>
> If there are any further issues or additional points you would like us to address, we are more than happy to provide any necessary explanations.
>
> Best Regards,
>
> The Authors

---

> ### Author Response · Authors · 2025-08-06
> **Second Follow-up to Reviewer 7Uo4**
>
> Dear Reviewer 7Uo4,
>
> Thank you again for your feedback. We have provided detailed clarifications and revision plan addressing all your concerns in previous Official Comments. We believe the issues you raised are well-addressable through our proposed revisions.
>
> If you have any additional questions about our clarifications or revision plan, we would be happy to address them. Looking forward to your continued engagement.
>
> Best regards,
>
> The Authors

---

> > ### Comment · Reviewer_7Uo4 · 2025-08-07
> >
> > Thank you for your response.
> >
> > As I outlined in my previous comments, the initial submission contains numerous misleading elements. However, I appreciate your revised plan and strongly hope you to thoroughly address all my concerns in the revised version. I will update my score accordingly.

---

> > > ### Author Response · Authors · 2025-08-07
> > >
> > > Dear Reviewer,
> > >
> > > Thank you for raising score. We will definitely incorporate all the clarifications and revisions outlined in the rebuttal to strengthen our paper. Your suggestions are invaluable in improving our paper.
> > >
> > > Warm Regards,
> > >
> > > The Authors

---

### Official Review · Reviewer_Xen8 · 2025-07-01

**Clarity:** 4
**Significance:** 3
**Originality:** 3
**Rating:** 5
**Confidence:** 4

**Summary:**

The paper presents MMaDA, an 8-billion-parameter multimodal large diffusion language model that unifies textual reasoning, multimodal understanding, and text-to-image generation inside a single discrete-diffusion Transformer.
Key ingredients are (i) a modality-agnostic diffusion backbone, (ii) Mixed Long-CoT supervised fine-tuning that supplies cross-modal chain-of-thought traces, and (iii) UniGRPO, a diffusion-friendly policy-gradient algorithm that combines KL regularisation, mask-ratio scheduling, and multiple modality-specific rewards in one objective.
Trained for three stages on 64 A100-80 GB GPUs (≈ 650 k steps), MMaDA achieves:

1. Multimodal understanding gains over strong baselines on POPE, VQAv2, MMMU, and other suites.
2. Best CLIP-Score 32.5 and ImageReward 1.15 among all unified models tested.
3. Competitive textual reasoning (e.g., GSM8K 73.4) while retaining diffusion’s parallel sampling; only 15–50 denoising steps suffice for good image quality.

**Questions:**

* Could you isolate each UniGRPO component—KL term, mask-ratio schedule, text vs image-reward heads?
* How sensitive is performance to the relative weights of correctness, CLIP, and ImageReward signals?
* If the curation LLM/VLM is weaker (e.g., in low-resource languages), does Mixed Long-CoT still improve alignment, or does it propagate errors? Any plans for model-generated self-CoT data to reduce this dependency?
* What are the trade-offs of retaining the MAGVIT-v2 VQ tokenizer versus adopting higher-resolution or learned continuous encoders, and how might that affect future MMaDA scaling?

**Ethical Concerns:**

["NO or VERY MINOR ethics concerns only"]

**Final Justification:**

The paper presents MMaDA, an 8-billion-parameter multimodal large diffusion language model that unifies textual reasoning, multimodal understanding, and text-to-image generation inside a single discrete-diffusion Transformer.

After the rebuttal, the authors have addressed my concerns successfully, and I give my final score as accept.

**Limitations:**

Please refer to the weaknesses.

**Paper Formatting Concerns:**

None.

**Quality:**

4

**Strengths And Weaknesses:**

# Strengths
* A single diffusion language model handles generation and reasoning across modalities, eliminating separate AR blocks or specialised heads.
* The model outperforms or matches state-of-the-art unified systems on eight vision–language and six language benchmarks, demonstrating versatility.
* Three clearly separated stages and an ablation (Table 6) show how Mixed Long-CoT and UniGRPO work.

# Weaknesses
* KL constraint, uniform-ratio mask scheduling, and per-modality reward heads are all active at once; hyperparameters are numerous and strongly coupled. Current ablations toggle only the whole UniGRPO block, leaving the individual contribution and stability of each reward unclear.
* The CoT traces are generated and filtered by existing LLM/VLMs (GPT 4.1), raising questions about scalability to domains or languages where such models underperform.
* Using a fixed 32×32 VQ grid (f = 16) may bottleneck high-resolution details and limit future scaling compared with more expressive continuous encoders.
* Because UniGRPO mixes correctness, CLIP score, ImageReward, and format rewards, improper balancing could harm certain downstream tasks; stability analysis across seeds/tasks is missing.

---

> ### Author Rebuttal · Authors · 2025-07-30
>
> We sincerely thank you for your time and efforts in reviewing our paper. We are glad to see your acknowledgement of our model's single framework across modalities, state-of-the-art unified performance, and the clear clarification of our three stages of training and thorough ablations. We provide responses for your concerns below:
>
> **Q1: Comprehensive ablation studies of UniGRPO process.**
>
> A1: Thank you for the insightful suggestion. We agree that a more fine-grained ablation of the UniGRPO components is important for understanding their individual contributions and interactions. While our primary contribution lies in the unified diffusion framework and three-stage training, UniGRPO is implemented with **minimal deviations from the original GRPO to ensure reproducibility.** We retain KL regularization, use a uniform-ratio mask scheduler, and adopt equal reward weights across modalities for clarity.
> In response to your suggestion, we have conducted a more detailed ablation study starting from our stage-2 checkpoint. We independently modified each of the following components: (1) removing the KL constraint, (2) replacing uniform-ratio masking with random masking, and (3) varying the weighting of the reward heads across modalities. Each variant was trained for 4,000 steps on 64 A100 GPUs, and we report the performance on the representative benchmarks for three tasks in the table below:
>
> **Table a: Ablations on Components of UniGRPO.**
> |Setting|KL|Mask|Reward Weighting|GSM8K|GeoQA|Image Reward|
> |-|-|-|-|-|-|-|
> |**Full UniGRPO (Default)**|✓|Uniform-ratio|1:1:1|66.5|19.5|0.95|
> |w/o KL Constraint|✗|Uniform-ratio|1:1:1|66.1|19.8|0.95|
> |Random Masking|✓|Random|1:1:1|63.4|17.9|0.87|
> |Text Rewards↑|✓|Uniform-ratio|3:1:1|67.2|17.4|0.86|
> |MMU Rewards↑|✓|Uniform-ratio|1:3:1|64.5|20.5|0.91|
> |T2I Rewards↑|✓|Uniform-ratio|1:1:3|63.8|16.8|0.96|
>
> From the results above, we observe the following key findings: (1) The KL constraint has **minimal impact on overall performance**. It brings negligible improvement in text generation and does not significantly affect the multimodal components, aligning with recent GRPO-based works such as DanceGRPO [1] (2) Using a uniform-ratio mask scheduler **consistently improves model performance,** by better simulating the progressive noise levels in diffusion training. Please kindly refer to our **Appendix E.2**, where we reported our training curves using a uniform masking scheduler, further validating these quantitative results. (3) Varying reward weights across modalities leads to task-specific performance trade-offs. We use 1:1:1 for balance and simplicity, deferring optimization to future work. We will incorporate these findings in the final version to provide a more comprehensive analysis of UniGRPO and its effectiveness in training MMaDA. Thank you again for your valuable feedback.
>
> **Q2:How sensitive is performance to the relative weights of correctness, CLIP, and ImageReward signals?**
>
> A2: We thank the reviewer for raising this important point. We further analyze modality-specific reward weights.
> 1. Correctness vs. Format Reward
> We first examine how the balance between correctness reward and format reward affects performance in text reasoning and multimodal reasoning tasks. We conduct UniGRPO training for 4,000 steps from the same stage-2 checkpoint, varying the ratio between correctness reward and format reward across **1:1, 4:1, and 10:1** settings over **3 random seeds**. As shown in the table below, the 4:1 setting offers the best trade-off, preserving both performance and output consistency, and best stability across seeds with reduced variance.
> **Table b: Ablations on Correctness vs Format Ratio of UniGRPO.**
> | Correctness:Format Ratio | GSM8K | MATH500 | GeoQA | Formatting Acc. |Std. |
> |------|-------|-----|--------|----------|----|
> | 1 : 1 | 65.1  | 26.1    | 14.0| **84.1%** |0.71%|
> | 4 : 1 (default) | **66.5** | **28.5** | **19.5**|83.6%  | **0.35%**|
> | 10 : 1 | 66.3  | **28.5**    |19.2| 73.9%    |0.64%|
>
> 2. For image generation, we varied the ratio between CLIP Score and ImageReward across 1:3, 1:1, and 3:1, also over **3 random seeds**, and reported our ablation results as below:
> **Table c: Ablations on CLIP vs ImageReward Ratio of UniGRPO.**
> |CLIP : ImageReward Ratio|CLIPScore(×0.1)|ImageReward|Avg.|Std.|
> |-|-|-|-|-|
> |1:3|3.01|**0.97**|1.99|0.62%|
> |1:1(default)|3.04|0.95|**2.00**|**0.58%**|
> |3:1|**3.05**|0.91|1.98|0.63%|
>
>    All three reward ratios consistently lead to performance improvements, indicating that the image generation branch is relatively robust to moderate changes in reward composition. Among them, the 1:1 setting achieves the most balanced results and exhibits the lowest variance across training seeds.
>
> **Q3: Adaptation for weaker LLM/VLM**
>
> A3: Thank you for this insightful question. While we use a strong LLM  to ensure high-quality reasoning, we would like to clarify that the **Mixed Long-CoT stage is model-agnostic**, and our framework is designed to be compatible with **any LLM/VLM capable of producing CoT traces**.
> To evaluate its scalability and robustness in lower-resource or domain-specific scenarios, we conducted additional experiments using two alternative curation models:
> (1) **LLaMA3-8B**, representing a relatively weaker and smaller model compared to GPT4.1 in low-resource settings. (2) **Qwen2.5-Coder-7B**, a domain-specialized model for code generation.
> We used these models to generate CoT traces for corresponding tasks (general reasoning or coding), and performed Mixed Long-CoT fine-tuning for 80,000 steps starting from our stage-2 checkpoint. Results are shown below:
>
> **Table d: Ablations on Different LLMs for CoT Traces Curation.**
> | CoT Traces | GSM8K (Math) | MATH500 (Math) | HumanEval (Code) | MMLU (General) |
> | ------ | ----- | ----- | ----- | ------- |
> | MMaDA-stage 1 | 17.4     | 4.2     | 15.9           | 42.1      |
> | Generated by GPT-4.1  | **58.5**     | **20.5**      | 25.4             | **46.7**      |
> | Generated by LLaMA3-8B         | 49.3        | 18.2           | 21.8             | 43.7           |
> | Generated by Qwen2.5-Coder-7B  | -            | -              | **33.5**         | -              |
>
> As seen, while the weaker model (LLaMA3-8B) leads to a moderate drop in absolute performance, it still enables **improvement of alignment over the baseline.** Meanwhile, the domain-specific model Qwen2.5-Coder-7B yields **stronger alignment in its respective coding domain**, highlighting the adaptability of our framework to different model sources and domains. We believe these results demonstrate that our training mechanism of MMaDA remains effective and extensible even in scenarios where state-of-the-art curation models are unavailable.
>
> **Q4: Any plans for model-generated self-CoT data?**.
> A4: Thank you for the insightful suggestion. Given MMaDA’s unified understanding and generation capabilities, it naturally supports self-generated CoT data and **self-evaluation.**.
> After training, the model can generate its own reasoning traces, which can be filtered via (1) external validation (e.g., GPT-4.1), or (2) internal consistency checks—for instance, generating a caption from an image and then regenerating the image from that caption, with CLIP similarity as a self-evaluation signal.
>
> This self-bootstrapping approach is inspired by HermesFlow [2], showing that unified models can iteratively improve via reinforcement and self-generated data. We view this as a promising future direction and plan to explore it in subsequent work.
>
> **Q5: trade-off of retaining the MAGVIT-v2 VQ tokenizer versus adopting higher-resolution or learned continuous encoders**.
>
> A5: Thank you for pointing out this important consideration. Indeed, the VQ tokenizer presents limitations, and we elaborate on the trade-offs below:
> 1. First, our primary goal is to establish a **unified diffusion backbone** as a proof-of-concept for future multimodal models. Using a fixed-resolution VQ tokenizer (f = 16) enables **faster convergence and more efficient training**, making it ideal for validating our framework and its effectiveness across modalities.
> 2. For compression f=16, it's actually very common in image generation, for example,  SANA[3] compresses by 32X, but is still able to generate high-quality 4K images, and not a limiting factor for visual quality in generative tasks.
> 3. For image understanding, we agree that continuous encoders may preserve finer visual details for understanding. Prior work, such as Show-o [4] demonstrates this advantage. However, we note two key challenges: (i) The performance gap is **partly due to data scale and training paradigm, not just architecture**—SigLIP benefits from massive image-text pretraining, while current VQ tokenizers are trained solely on reconstruction. (ii) More importantly, using continuous encoders for understanding but discrete encoders for generation introduces a **modality gap in visual representations**, potentially degrading consistency in tasks like **multi-turn dialogue.**
>
>     To address this, we plan to explore unified discrete tokenizers such as UniTok [5], which integrate alignment objectives during tokenizer training. This direction offers a promising path toward consistent, scalable multimodal representation learning beyond MAGVIT-v2.
>
> [1] DanceGRPO: Unleashing GRPO on Visual Generation.
> [2] Hermesflow: Seamlessly closing the gap in multimodal understanding and generation.
> [3] Sana: Efficient high-resolution image synthesis with linear diffusion transformers.
> [4] Show-o: One single transformer to unify multimodal understanding and generation.
> [5] Unitok: A unified tokenizer for visual generation and understanding.

---

> > ### Comment · Reviewer_Xen8 · 2025-08-07
> >
> > Thanks for the response. My concerns have been addressed. I will raise my score to 5.

---

> > > ### Author Response · Authors · 2025-08-07
> > >
> > > Dear Reviewer,
> > >
> > > Thank you for raising score! Your feedback has been invaluable in improving our paper.
> > >
> > > Warm Regards,
> > >
> > > The Authors

---

### Official Review · Reviewer_gskN · 2025-07-02

**Clarity:** 3
**Significance:** 3
**Originality:** 3
**Rating:** 5
**Confidence:** 2

**Summary:**

This paper introduces MMaDA, a unified framework that bridges pretraining and post-training for diffusion-based multimodal foundation models. The work addresses a critical gap in existing unified multimodal models, which primarily focus on architectural design and pretraining strategies while neglecting post-training methodologies, particularly in non-autoregressive settings.

The paper makes three main contributions: (1) a unified diffusion architecture that eliminates modality-specific components by using discrete tokenization and masked token prediction for both text and images, (2) a mixed long CoT finetuning strategy that aligns reasoning across modalities using a unified format, and (3) UniGRPO, a novel reinforcement learning algorithm adapted for diffusion models that addresses specific challenges like local masking dependency and non-autoregressive sequence-level likelihoods. The model demonstrates competitive performance across textual reasoning, multimodal understanding, and text-to-image generation tasks.

**Questions:**

Given that the model is limited to 8B parameters, how is the approach expected to scale to larger model sizes (20B+)? Are there any fundamental barriers to scaling? Also, are there some insights or conditions where MMaDA performs poorly compared to other models? Please also refer to weakness.

**Ethical Concerns:**

["NO or VERY MINOR ethics concerns only"]

**Final Justification:**

The paper received a high score from me, and I’m keeping it as is.

**Limitations:**

yes

**Quality:**

3

**Strengths And Weaknesses:**

Strengths:

1. The adaptation of GRPO to diffusion models is technically sound and addresses challenges in applying RL to non-autoregressive models.

2. The unified probabilistic formulation using discrete diffusion for both text and images is well-motivated.

3. Strong empirical results across multiple domains, achieving state-of-the-art performance on several benchmarks while maintaining competitive results on others.

4. The demonstration that diffusion models can serve as general-purpose language models is significant.

Weakness:

1. The paper lacks comprehensive ablation studies to isolate the contribution of each component (unified architecture vs. mixed CoT vs. UniGRPO). With three major innovations introduced simultaneously, it's unclear which components are most critical for the observed performance gains and which might be incremental improvements. Is it possible to provide systematic ablation experiments that isolate the individual contributions of the unified diffusion architecture, the mixed long-CoT finetuning, and the UniGRPO algorithm, in some creative experimental design?

---

> ### Author Rebuttal · Authors · 2025-07-30
>
> We sincerely thank you for your thorough evaluation and constructive feedback. We are grateful for your recognizing the technical soundness of our UniGRPO method, the motivation of our unified discrete formulation across modalities, the strength of our empirical results, and the significance of our research direction. We address the specific comments and questions below.
>
> **Q1:Lack of comprehensive ablation studies**
>
> A1: We thank the reviewer for this insightful suggestion. We agree that a comprehensive ablation study is crucial for understanding the individual contributions of our key components. However, conducting a full factorial ablation on an 8B-parameter model is computationally prohibitive. Within the limited time of the rebuttal period, we have performed as many ablations as possible and report preliminary results to isolate the contributions of each major component under constrained training budgets.
>
> 1. **Ablations of the part of the unified architecture.**
> To examine the core contribution of MMaDA’s unified probabilistic formulation based on discrete diffusion, we conducted a comparison experiment with an autoregressive (AR) baseline using LLaMA-3 8B as the backbone. **The only difference lies in the training loss and inference method: the AR baseline uses AR-style likelihood loss and greedy sampling, while MMaDA employs diffusion-based objectives and decoding.** This baseline resembles Emu3 [1], an AR-based unified multimodal model.
> Both models were trained under identical settings (data, compute, batch size) for **100K** steps using 64 A100 GPUs. For both pre-trained models, we evaluate on foundational generative tasks: conditional text generation using Perplexity (PPL), image captioning using CLIP Score, and class-conditioned image generation using FID on ImageNet. We also report image generation latency to assess practical efficiency. The results are reported as follows:
> **Table a: Comparison with AR architectures**
> | Model                     | PPL ↓ | Image-Caption CLIP Score ↑ | FID ↓   | Image Generation Latency↓ |
> | ------------------------- | ---- | -------------------------- | ------- |-|
> | AR (LLaMA-3 8B, AR loss)  | **23.1** | 14.5                       | 16.2    |~120s|
> | MMaDA (Unified Diffusion) | 25.3 | **16.9**                   | **12.4** | **~7s**|
>
>     These results show that, under matched conditions, MMaDA **outperforms the AR-based baseline on image generation and multimodal understanding tasks**, while maintaining competitive performance on language modeling (PPL). Most notably, MMaDA achieves a **17× speedup in image generation latency**, highlighting the efficiency advantage of diffusion-based decoding. The improvements stem from the structural benefits of diffusion models for vision tasks: diffusion avoids the **unnatural raster-order decoding and error accumulation** observed in AR models, which results in both lower fidelity and substantially higher latency. While the AR baseline achieves slightly better PPL, we attribute this to the relatively underexplored decoding strategies for diffusion-based language generation.
>     To further validate this conclusion, we note that a recent work, **UniDisc [2]**, independently explores unified modeling using discrete diffusion and compares it with autoregressive (AR) approaches on joint text-image generation and inpainting tasks. Their findings are consistent with ours: discrete diffusion models offer **substantial advantages in image generation**, while achieving **comparable performance in text generation**. These aligned observations across independent studies strengthen our claim that discrete diffusion serves as a more natural and effective generative mechanism for multimodal modeling compared to AR-based approaches.
>
> 2. **Ablations on mixed long-CoT fine-tuning, and the UniGRPO**
> For these two parts of MMaDA, we included our evaluation results of MMaDA across stages in **Appendix E.1**. As shown in Table 6, we report model performance after each training stage across a range of reasoning and generation benchmarks. After Mixed Long-CoT fine-tuning, the model shows substantial gains in tasks requiring multi-step reasoning, particularly in mathematics (GSM8K: +47.8, MATH500: +22.3) and geometric understanding. UniGRPO further boosts performance across all benchmarks. These results clearly demonstrate the complementary benefits of each component: Mixed Long-CoT fine-tuning primarily enhances reasoning abilities through multi-hop instruction tuning, while UniGRPO improves both understanding and generation through reinforcement-based optimization. Together, they enable our model to match or exceed the performance of strong task-specific baselines in a unified manner.
>
>
> **Q2: Scalabilities of our model?**
>
> A2: The principle that performance scales with model size is well-documented for autoregressive LLMs. Encouragingly, this trend appears to hold for diffusion-based and unified models as well. The performance leap from MDLM [3] to LLaDA [4] in diffusion LLMs, and the significant gains observed when scaling from Janus-1.5B [5] to Janus-pro-7B [6] in unified MLLMs, strongly suggest that our framework would similarly benefit from larger model sizes.
>
> **Q3: Fundamental barriers of scaling?**
>
> A3: The primary barrier to scaling at present is not architectural but rather the availability of **powerful, open-source base diffusion language models.** Our unified probabilistic formulation requires a diffusion-based text model as its foundation. The current open-source ecosystem is largely limited to models in the ~8B parameter range (e.g., LLaDA-8B [4], Dream-7B [7]). The future development of larger and more capable base diffusion LLMs would directly enable us to scale our training paradigm and likely unlock substantial further performance gains.
>
> **Q4: Conditions for Poor Performance?**
>
> A4: Our model currently faces challenges in tasks that demand extremely **fine-grained visual understanding**, as well as OCR tasks.
> This limitation stems from two main factors related to our vision encoder. We use a **MAGViT-v2-based VQ encoder.** While effective for general scene comprehension, its reconstructive fidelity for intricate patterns like text is limited. The publicly available weights are not sufficiently trained for these specialized, fine-grained tasks.
>
> Besides, the visual encoder was primarily trained on a reconstruction objective. This is a gap compared to state-of-the-art approaches that use vision encoders like SigLIP, which are pre-trained on massive datasets for deep semantic understanding.
> We identify this as a key area for future work. In the future, we plan to integrate a more powerful and **unified tokenizer**, such as UniTok[8], into our framework.  We believe this will significantly enhance the model's perceptual capabilities.
>
> [1] Emu3: Next-token prediction is all you need
> [2] Unified multimodal discrete diffusion
> [3] Simple and effective masked diffusion language models. Neurips 2024
> [4] Large language diffusion models.
> [5] Janus: Decoupling visual encoding for unified multimodal understanding and generation. CVPR 2025.
> [6] Janus-pro: Unified multimodal understanding and generation with data and model scaling.
> [7] Dream-7B.
> [8] Unitok: A unified tokenizer for visual generation and understanding.

---

### Decision · Program_Chairs · 2025-09-17

**Decision:**

Accept (poster)

**Comment:**

The paper received mixed ratings from the reviewers. Most of the reviewers are positive about the submission, in terms of the technical contribution and the experimental evaluation. One reviewer raised several questions regarding the presentation and experimental details. These issues have been well addressed in the rebuttal and have been acknowledged by the reviewer. As the overall tone from the reviewers is supportive, AC finally decided to recommend acceptance of the submission.